# Broadband high-Q multimode silicon concentric racetrack resonators for widely tunable Raman lasers

Yaojing Zhang [1,2✉], Keyi Zhong[1,2], Xuetong Zhou [1] & Hon Ki Tsang [1✉]

Multimode silicon resonators with ultralow propagation losses for ultrahigh quality (Q) factors have been attracting attention recently. However, conventional multimode silicon resonators only have high Q factors at certain wavelengths because the Q factors are reduced at wavelengths where fundamental modes and higher-order modes are both near resonances. Here, by implementing a broadband pulley directional coupler and concentric racetracks, we present a broadband high-Q multimode silicon resonator with average loaded Q factors of $1.4 \times 10^6$ over a wavelength range of 440 nm (1240–1680 nm). The mutual coupling between the two multimode racetracks can lead to two supermodes that mitigate the reduction in Q factors caused by the mode coupling of the higher-order modes. Based on the broadband high-Q multimode resonator, we experimentally demonstrated a broadly tunable Raman silicon laser with over 516 nm wavelength tuning range (1325–1841 nm), a threshold power of $(0.4 \pm 0.1)$ mW and a slope efficiency of $(8.5 \pm 1.5)$ % at 25 V reverse bias.

[1] Department of Electronic Engineering, The Chinese University of Hong Kong, Shatin, New Territories, Hong Kong. [2]These authors contributed equally: Yaojing Zhang, Keyi Zhong. ✉email: yaojingzhang@cuhk.edu.hk; hktsang@ee.cuhk.edu.hk

Resonators with high quality (Q) factors can build up large intensities at resonances, enabling efficient optical pumping of nonlinear optical devices[1]. Raman lasers based on stimulated Raman scattering (SRS) in high-Q resonators can attain quite low pump threshold powers and high Stokes output powers if the cavities were designed to have high Q factors at both pump and Stokes wavelengths[2–6]. SRS is mediated by a coherent phonon population and does not require phase matching between the pump and Stokes waves. Raman lasers are therefore useful to produce output wavelengths at longer wavelengths than available from the pump laser[7–23]. Carefully engineering the resonators to have high Q factors over a broad wavelength range is necessary if the Raman lasers were to have a wide tuning range at every resonant wavelength[24]. Widely tunable Raman lasers have been well demonstrated in diamond[25–28], silica[29], chalcogenide glass[30,31], and aluminum nitride[32] devices. A wide tuning bandwidth of about 100 nm was realized in a single-mode racetrack microresonator fabricated on a diamond waveguide platform with a lasing threshold power of 85 mW[33]. More recently, Raman lasing was tuned from 1615 nm to 1755 nm together with its cascaded modes in a single-mode $Ge_{25}Sb_{10}S_{65}$ resonator with a pump threshold of 3.25 mW[31]. In silicon, the Stokes wave has a 15.6 THz frequency detuning from the pump wave. A tuning range of 83 nm with a threshold pump power of 15 mW was reported recently in a reverse-biased single-mode silicon racetrack resonator[34].

Single-mode waveguide resonators have good mode confinements for enhanced optical nonlinearities[35,36]. However, high-index-contrast single-mode silicon waveguide-based resonators fabricated by commercial multi-project wafer foundries suffer from high propagation losses of typically about 2 dB/cm, mainly produced by the scattering losses which are associated with sidewall roughness and surface defects. The highest Q factors of these resonators[37] are usually limited to around $10^5$. Further fabrication processes can be used to reduce the losses of the single-mode waveguides[38–41]. Using wider, multimode, waveguides offer smaller propagation losses for the fundamental modes because of the reduced modal overlap with the sidewall roughness. Multimode waveguide racetrack resonators[42,43] have routinely achieved loaded Q factors of well over $10^6$. Conventional multimode racetrack resonators consist of one multimode bus waveguide and one multimode racetrack. In such multimode racetrack resonators, the fundamental mode can couple to the higher-order mode in the multimode bends, leading to the reduction in Q factors of the fundamental mode-based resonances[44–46]. This is a well-known problem and has been previously tackled by using single-mode bends[47,48], Euler bends[42,43,49], and Bezier bends[50] to suppress the higher-order modes at the bends and maintain a high Q factor for the fundamental mode.

Here, we propose a new approach using multimode concentric racetracks together with a broadband pulley design on the directional coupler region to maintain ultrahigh Q factors over a broadband wavelength range in the multimode silicon resonator as shown in Fig. 1a. Even though higher-order modes could be generated at the racetrack bends, the mutual coupling of the two multimode racetracks enables the individual fundamental modes of the racetracks to form anti-symmetric and symmetric supermodes of the coupled waveguide system. The two supermodes have high loaded Q factors but with slightly different resonant frequencies, and their detuning alleviates the effects of the coupling to higher-order modes at the fundamental mode resonances and ensures there is at least one high-Q resonance of the resonator in each free spectral range (FSR). Experimental results validate that the resonator achieves consistently high Q factors, averaging at about $1.4 \times 10^6$ over the wavelength range from 1240 to 1680 nm. The multimode concentric resonator well ameliorates the problem of localized regions where the frequency spectrum

has large decreases in Q factors arising from the degeneracy of resonant frequencies at the mode coupling regions in multimode resonators. The multimode concentric resonator allows high Q factors to be maintained in every single FSR over a broad optical bandwidth. Based on this resonator, we experimentally demonstrated a widely tunable Raman lasing spanning from 1325 to 1841 nm. The average lasing threshold pump power and lasing slope efficiency were measured as $(0.4 \pm 0.1)$ mW and $(8.5 \pm 1.5)\%$ at 25 V reverse bias.

## Results

**Broadband pulley directional coupler design.** Here, we used the well-established concept of a pulley directional coupler[51] to achieve a broadband operation of the resonator. To reduce the wavelength dependence of the coupler, the outer racetrack and the bus waveguide have different widths and bending radii, which are carefully chosen to introduce sufficient phase mismatch. With the fixed width and bending radius of the outer racetrack, the widths of the bus waveguide and the gap are adjusted. The designs are detailed in Section 7 of the Supplementary Information. The coupling region is formed by the outer racetrack with 1.5 μm width and 200 μm radius, the bus waveguide with 0.88 μm width, and a 400 nm gap between the outer racetrack and the bus waveguide. We first calculated the variation of coupling fractions of the directional coupler with wavelengths changing from 1200 to 1700 nm. The total coupling fraction fluctuates about an average value of 2.82%. The coupling fraction from the bus waveguide to the TE0 mode of the outer racetrack has a 1 dB bandwidth of about 283 nm at wavelengths from 1300 to 1583 nm. The corresponding coupling Q factors[52] can be calculated with an average value of $2.3 \times 10^6$ in Fig. 1b.

**Broadband high-Q multimode concentric racetrack resonator design.** The resonator is comprised of two multimode concentric racetracks, which are coupled to each other, and the outer racetrack is coupled to a bus waveguide. The geometric parameters of the bus waveguide ($w_0$), the outer ring width ($w_{out}$), the inner ring width ($w_{in}$), the gap between the bus waveguide and the outer ring (gap1), the gap between the inner and outer racetracks (gap2), the bending radius of the outer ring ($R_{out}$), the length of the straight section ($L_s$), the waveguide height, and the etching depth are designed as 0.88 μm, 1.5 μm, 3 μm, 0.4 μm, 0.4 μm, 200 μm, 690 μm, 220 nm, and 70 nm, respectively. With the mutual coupling between the two racetracks, the light propagating in the bus waveguide would couple into the outer racetrack first, and then into the inner racetrack. The gap between the bus waveguide and the inner racetrack is designed to be sufficiently large to ensure that there is a negligible direct coupling between them. With the geometric parameters, we first simulate the excited eigenstates of the supermodes in the concentric racetrack resonator in Fig. 1c. The eigenmode in Fig. 1c propagates mainly in the outer racetrack with a modal profile similar to the TE0 mode of the outer racetrack, labeled as $Out_{TE0}$. Similarly, the other two modes in Fig. 1c are labeled as $In_{TE0}$ and $In_{TE1}$. Both outer and inner racetrack resonators are multimode indicated by the existence of higher-order modes in the simulated mode profiles in Fig. S10 (see Section 4 of the Supplementary Information).

In concentric racetrack resonators, the outer racetrack has a longer roundtrip length than the inner racetrack. To get equal optical path length (OPL) for the fundamental modes supported by the outer and inner racetracks without coupling to satisfy the phase-matching condition, the refractive index of the fundamental mode of the outer racetrack must be smaller than that of the inner racetrack[53]. Therefore, the width of the outer racetrack

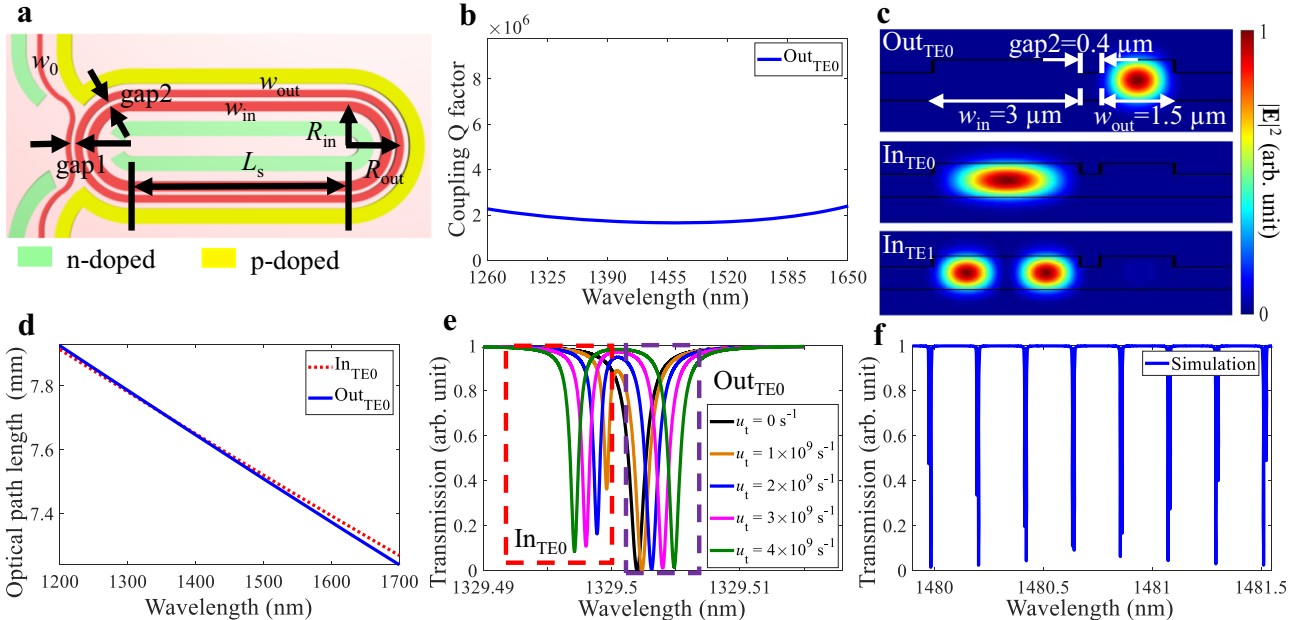

**Fig. 1 The concentric racetrack resonator design. a** Structure of the concentric racetrack resonator with a p-i-n junction to remove the free carriers. Schematic from the top view with geometric parameters. $w_O$: 0.88 μm; $w_{in}$: 3 μm; $w_{out}$: 1.5 μm; gap1: 0.4 μm; gap2: 0.4 μm; $R_{out}$: 200 μm; $L_s$: 690 μm. **b** Simulated coupling Q factors versus wavelengths from 1200 to 1700 nm. **c** Simulated mode propagates mainly in the outer racetrack, and modes are excited in the inner racetrack. The color bar indicates the intensity of the electric field |**E**|. **d** Calculated optical path lengths at the fundamental transverse electric (TE0) modes of the outer and the inner racetracks. **e** Simulated transmission spectra around 1329.5 nm with different coupling coefficients $u_t$ between the inner and outer racetracks. **f** Simulated transmission spectrum of the multimode concentric resonator from 1480 to 1481.5 nm.

must be narrower than the inner racetrack. The OPL can be calculated by,

$$OPL_{out/in} = 2L_{s,out/in}n_{s,out/in} + 2\pi R_{out/in}n_{bend,out/in}, \quad (1)$$

$n_{s, out/in}$ and $n_{bend, out/in}$ are the effective indices of the straight and bending sections at the racetracks. For our design, the widths of the inner and outer racetracks are set to match the equal OPL at the wavelength around 1330 nm in Fig. 1d. With the equal OPL, the two modes can couple to each other after one roundtrip at 1330 nm. At other wavelengths, even though the two modes have different OPLs, mutual coupling between them can still happen after several roundtrips with an overall balance between the two modes to have the same accumulated roundtrip phase delays[53].

The concentric racetracks can be modeled using equations[54,55]:

$$\frac{dA_{out}}{dt} = \left(j\omega_{out} - \frac{1}{\tau_{out}}\right)A_{out} - j\kappa S_{in} - ju_t A_{in}, \quad (2a)$$

$$\frac{dA_{in}}{dt} = \left(j\omega_{in} - \frac{1}{\tau_{in}}\right)A_{in} - ju_t A_{out}, \quad (2b)$$

$$S_{out} = e^{-j\phi}(S_{in} - j\kappa^* A_{out}), \quad (2c)$$

where $A_{in}$ and $A_{out}$ are the amplitudes of modes propagating in the inner and outer racetracks; $\omega_{in}$ and $\omega_{out}$ are the resonant frequencies of the inner and outer racetracks without mutual coupling; $\tau$ is the photon lifetime; $\kappa$ is the coupling coefficient between the bus waveguide and the outer racetrack; $u_t$ is the coupling coefficient in the time domain between the inner and outer racetracks; $S_{in}$ and $S_{out}$ are the amplitudes of the input and output pump waves; $\phi$ is the phase change from the input to output states. When $u_t$ cannot be neglected or $\Delta\omega$ is small enough, mode splitting induced double resonances would occur in one FSR[54,56–58]. Because the inner resonator does not directly interact with the bus waveguide, $In_{TE0}$ does not couple directly to the bus waveguide. The coupling only occurs between $Out_{TE0}$ and

$In_{TE0}$. When $Out_{TE0}$ and $In_{TE0}$ couple with each other, they mainly share the propagation loss. We assume the propagation losses of the outer and inner racetracks are 0.4 and 0.2 dB/cm[43]. Considering the $Out_{TE0}$ and $In_{TE0}$ with parameters listed in Table S3 (see Section 2 of the Supplementary Information), we solved the above equations and obtained the transmission spectra of the multimode concentric resonator with different mutual coupling coefficients $u_t$ as shown in Fig. 1e. The mode splitting generates anti-symmetric and symmetric modes. The two modes produced from the mutual coupling propagate along the optical paths between the two racetracks. They satisfy[53,59–61]:

$$\omega_{as/s} = \frac{\omega_{in} + \omega_{out}}{2} + j\left(\frac{1}{\tau_{in}} + \frac{1}{\tau_{out}}\right)/2$$
$$\pm\sqrt{\left[\frac{\omega_{in} - \omega_{out}}{2} + j\left(\frac{1}{\tau_{in}} - \frac{1}{\tau_{out}}\right)/2\right]^2 + u_t^2}. \quad (3)$$

'+/−' sign donates for $\omega_{as}/\omega_s$ of the anti-symmetric/symmetric mode.

$$S_{out} = \hat{t}_{bus,out}S_{in} - j\hat{\kappa}_{bus,out}A_{out}, z \in [0, L_c] \quad (4a)$$

$$A_{out1} = -j\hat{\kappa}_{bus,out}S_{in} + \hat{t}_{out}A_{out} - j\hat{\kappa}_{out,in}A_{in}, z \in [0, L_c] \quad (4b)$$

$$A_{in1} = -j\hat{\kappa}_{out,in}A_{out} + \hat{t}_{out,in}A_{in}, z \in [0, L_c] \quad (4c)$$

$$\frac{\partial A_{out1}}{\partial z} = (-j\beta_{out} - \frac{\alpha_{out}}{2})A_{out1} - ju_L A_{in1}, z \in [L_c, L_c + L_{mc}] \quad (4d)$$

$$\frac{\partial A_{in1}}{\partial z} = (-j\beta_{in} - \frac{\alpha_{in}}{2})A_{in1} - ju_L A_{out1}, z \in [L_c, L_c + L_{mc}] \quad (4e)$$

$$A_{out} = A_{out2}e^{-(j\beta_{out} + \frac{\alpha_{out}}{2})(L_{out} - L_{mc} - L_c)}, z \in [L_c + L_{mc}, L_{out}] \quad (4f)$$

$$A_{in} = A_{in2}e^{-(j\beta_{in} + \frac{\alpha_{in}}{2})(L_{in} - L_{mc} - L_c)}, z \in [L_c + L_{mc}, L_{in}] \quad (4g)$$

Besides the above analyses in the time domain, the dynamics of concentric resonators can also be explained in the spatial domain[62,63] using Eqs. (4a–4g). $A_{in}$ and $A_{out}$, $A_{in1}$ and $A_{out1}$, $A_{in2}$ and $A_{out2}$ are the amplitudes of modes propagating mainly in the inner and outer racetracks before and after coupling with bus waveguide as well as mutual coupling, respectively. $\hat{t}_{bus,out}/\hat{\kappa}_{bus,out}$ and $\hat{t}_{out,in}/\hat{\kappa}_{out,in}$ are the amplitude transmission/coupling coefficients between the bus waveguide and outer resonator, and between the outer resonator and inner resonator[64]. $\hat{t}_{out}$ is the amplitude of transmission coefficient of the outer resonator and set to be $\sqrt{1 - \hat{\kappa}_{bus,out}^2 - \hat{\kappa}_{out,in}^2}$. $u_L$ is the mutual coupling coefficient in the spatial domain. $\alpha$ and $\beta$ are the propagation loss and constant. $L_c$ is the coupling length between the bus waveguide and resonator. $L_{mc}$ is the mutual coupling length. Equations (4a–4c) depict the variations of the electric field amplitudes of the bus waveguide, the outer and inner resonators in the coupling region, $[0, L_c]$. We separate the mutual coupling [Eqs. (4d, 4e)] and the pure propagation [Eqs. (4f, 4g)], since the roundtrip lengths of inner and outer resonators ($L_{in}$ and $L_{out}$) are different. In simulation, we set $L_c = 0$ and $L_{mc} = L_{in}$ for simplification. By solving the above equations, we obtained the simulated transmission in Fig. 1f, both split two supermodes from the mutual coupling between $Out_{TE0}$ and $In_{TE0}$ have loaded Q factors over $10^6$.

In principle, when the resonant frequencies of the fundamental modes of inner and outer resonators approach each other, mutual coupling enables mode splitting to form symmetric and anti-symmetric modes. One mode propagates mainly in the outer resonator, while the other is mainly in the inner resonator. Because the inner resonator is designed to have a larger width than the outer resonator, the mode in the inner resonator has a smaller propagation loss. In this scenario, the mutual coupling between the two resonators slightly reduced the effective propagation loss of the mode in the outer resonator, while the mutual coupling can slightly increase the effective loss in the inner resonator. Since both modes have low propagation losses, after the mutual coupling, two comparable high-Q resonances could generate within one FSR. Even if one high-Q resonance couples with a higher-order mode leading to a seriously reduced loaded Q factor, there remains the other resonance to maintain one high-Q resonance in the same FSR.

The multimode concentric racetrack resonator embedded with a p-i-n junction in Fig. 2a is fabricated on a 220 nm-hick silicon-on-insulator wafer with the geometric parameters listed above. The width between the p-doped and n-doped regions is 8.9 μm. In the measurement, one O-band tunable laser TSL 550 (SANTEC corporation) and one C-band tunable laser Keysight 81608 A (Keysight Technologies corporation) with input powers of 0.1 mW are separately conducted to characterize the transmission spectra from 1260 to 1360 nm and from 1450 to 1650 nm in Fig. 2b, c. The insertion loss of fiber-chip-fiber at 1550 nm is 8.2 dB. The mode splitting from the strong mutual coupling between the inner and outer racetracks occurs around 1329.5 nm in Fig. 2d, in accord with the designed 1330 nm. The generated two high-Q resonances within one FSR ensure that at least one high-Q resonance can be used even if the other one is impaired by the higher-order mode. A highly decreased Q factor would happen under the condition that both fundamental and higher-order modes are on resonance, and if mode splitting were not present. For our device, highly decreased Q factors do not happen from 1240 to 1680 nm.

Besides, apart from the 1330 nm, mode splitting also arises at other wavelengths where the accumulated roundtrip phase delays are equal, for example, in the transmission spectrum

(Fig. 2e) around 1480 nm. From Fig. 2e, the higher-order mode (inverted triangle) slightly couples to the right TE0 resonance, while the left TE0 resonance is barely affected. The experimental spectrum in Fig. 2e) agreed well with the simulation in Fig. 1f). The above models and experiments well prove that the multimode concentric resonator can help to mitigate the reduction in loaded Q factor due to the influence of higher-order modes on the TE0 modes by introducing two high-Q resonance modes within one FSR induced by the interaction between the $Out_{TE0}$ and $In_{TE0}$. On the contrary, in a multimode single resonator, the highly decreased loaded Q factors induced by the mode coupling from higher-order modes can be up reduced by 81%. The corresponding theoretical models and experimental results are included in Sections 1–3 of the Supplementary Information.

The intrinsic Q ($Q_i$), coupling Q ($Q_c$), and loaded Q ($Q_L$) factors of the TE0 modes at resonant wavelengths from 1260 to 1360 nm and from 1450 to 1650 nm in each FSR are shown in Fig. 2f. Both $Q_c$ and $Q_L$ factors exhibit relatively flat variations from 1260 to 1650 nm, indicating the well-designed pulley directional coupler for broadband resonator-waveguide coupling[52,65,66]. We do not have measurements from 1360 to 1450 nm because of the limited wavelength coverage of pump lasers in our laboratory. Figure 1b shows the simulation results of the directional coupler, in which the $Q_c$ factors from 1360 to 1450 nm maintain comparable values, which is confirmed by the measurement in Fig. 2f. The measured results show that the multimode concentric resonator maintains high Q factors over the wavelength range from 1260 to 1650 nm, with average $Q_L$ and $Q_i$ factors of $1.4 \times 10^6$ and $3.7 \times 10^6$. The average propagation loss of the resonator is calculated to be $0.3 \pm 0.15$ dB/cm.

**Broadband tunable Raman silicon laser**. The optical phonon frequency in silicon resonates at 15.6 THz and produces a narrowband Stokes shift corresponding to this frequency. We first measure the Stokes output power (after being coupled out to the bus waveguide) as a function of pump power (in the bus waveguide before coupling into the resonator) to get the lasing threshold power at the pump wavelength of 1550 nm shown in Fig. 3. The experimental setup is detailed in Section 6 of the Supplementary Information. Pumping with quasi-transverse-electric polarization at 1550 nm wavelength under reverse biases of 25 and 10 V, we start to observe the lasing at coupled pump powers of 0.3 and 0.4 mW. The threshold densities are 833 and 1111 W/mm². After the lasing threshold powers, the laser output powers increase rapidly to the maximum values of 0.02 and $0.6 \times 10^{-3}$ mW at coupled pump powers of 1.3 and 1 mW, followed by the decrease in output powers and tending to be saturated finally. The initial slope efficiency (at 25 V bias) was measured as 10% from the linear fit curve. The Stokes lasing power is enhanced by applying the 25 V reverse bias. Since a further increase of the reverse bias did not significantly increase the photocurrent (Section 9, Supplementary Information), we used the 25 V bias in our measurements. Besides, at a larger bias than 25 V, catastrophic breakdown damage occurred in some devices. Including the lasing threshold power of 0.5 mW and slope efficiency of 7% at the mode coupling region (see Fig. S12, Supplementary Information), the average lasing threshold power and slope efficiency of this resonator were measured as $(0.4 \pm 0.1)$ mW and $(8.5 \pm 1.5)$%. The experimental results agreed well with the theoretical predictions in Section 5 of the Supplementary Information.

To characterize the Stokes Raman lasing output with different pump wavelengths, we sought to use the same coupled pump power at each pump wavelength to allow comparison of the Stokes output

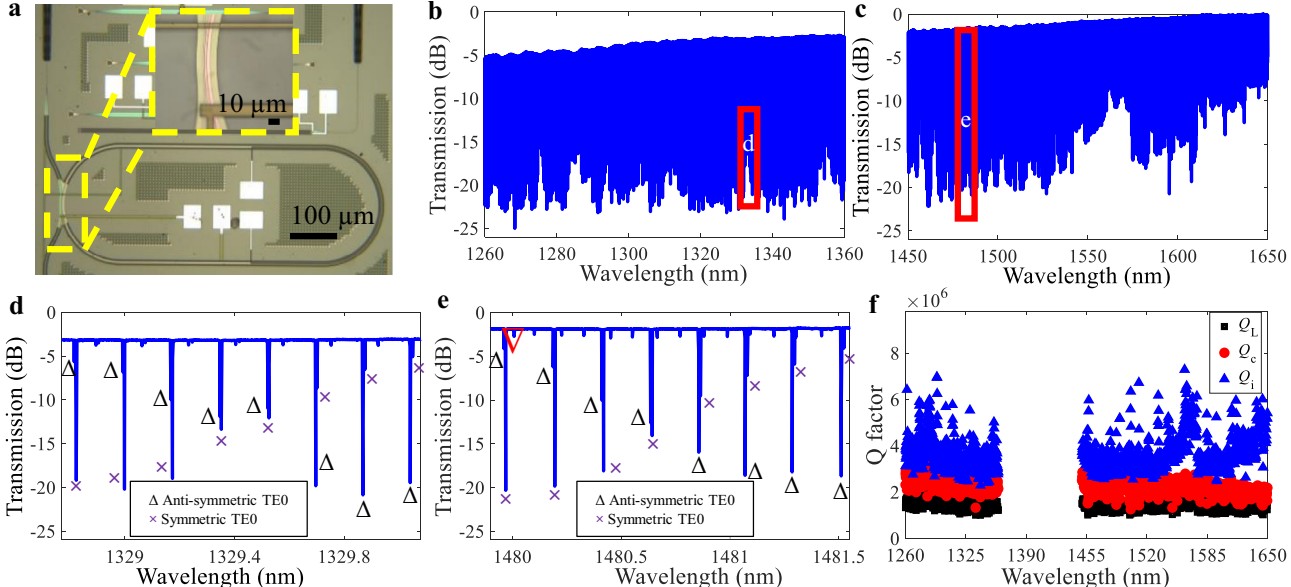

**Fig. 2 Transmission characteristics of the concentric racetrack resonator. a** Microscope image of the concentric resonator with zoom-in directional coupler region in the inset. **b**, **c** Transmission spectra with wavelength ranges from 1260 to 1360 nm and from 1450 to 1650 nm. **d**, **e** Selected transmission spectra region around 1329.5 and 1480.5 nm where the inverted triangle marks the higher-order mode. **f** Intrinsic Q ($Q_i$), coupling Q ($Q_c$), and loaded Q ($Q_L$) factors of the TE0 modes at resonant wavelengths from 1260 to 1360 nm and from 1450 to 1650 nm (the absent wavelength range covering 1360–1450 nm is limited by the tunable lasers).

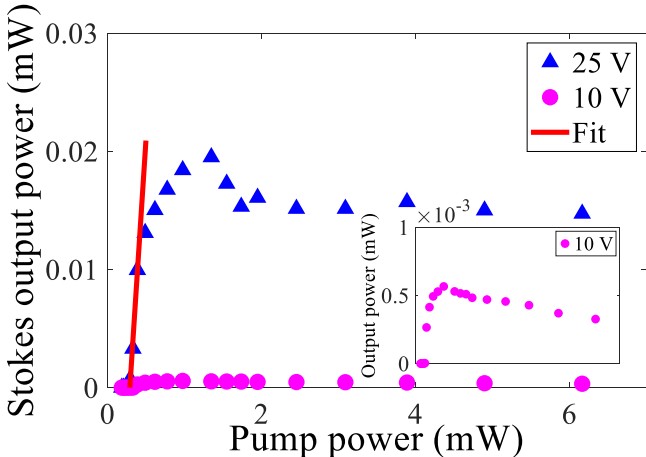

**Fig. 3 Stokes Raman laser output power as a function of pump power at pump wavelength of 1550 nm with 25 and 10 V reverse biases.** The inset is the zoom-in image for the 10 V reverse bias. The linear fitting curve shows the slope efficiency to be 10% at 25 V reverse bias.

power as a function of the pump wavelength. Actually, in the measurement, the tunable laser has different maximum powers at different wavelengths and coupling losses are also not constant at different wavelengths. We took into account the coupling loss to get similar coupled pump power at each pump wavelength. Here, the coupled pump power is the pump power coupled into the bus waveguide. We tuned the pump wavelength from 1240 to 1380 nm using an O-band tunable laser Keysight 81606A (Keysight Technologies corporation) and 1450 to 1680 nm using a C-band tunable laser Keysight 81608A (Keysight Technologies corporation) and an L-band tunable laser TSL 510 (SANTEC corporation). The wavelength step is 10 nm. We injected the same coupled pump powers of 1.3 mW at these wavelengths as shown in Fig. 4a, b. The corresponding Stokes

laser output powers are listed in Fig. 4a, b. The laser output power is the output power before being coupled out of the edge coupler. We show the lasing spectra of 1325.5, 1359.9, 1394.6, 1428.8, 1463.4, 1486.5, 1568.8, 1627.4, 1686, 1745.6, 1804.6, and 1840.7 nm at pump wavelengths of 1239.9, 1270.1, 1300.2, 1329.9, 1359.9, 1379.9, 1450.4, 1500.3, 1550, 1600.2, 1650.2, and 1680 nm in Fig. 4c, respectively. Two optical spectrum analyzers of Yokogawa AQ6370D (600–1700 nm) with a resolution of 0.02 nm and Yokogawa AQ6375B (1200–2400 nm) with a resolution of 0.05 nm were used to record the lasing spectra up to 1841 nm. The linewidth of the Raman laser was measured by over two orders narrower than the resonator linewidth in two independent experimental measurements based on self-homodyne measurement with a delay interferometer and heterodyne linewidth measurement using a second tunable laser[67].

Generally, the laser output power decreases with the increase of the pump wavelength because the Raman gain coefficient is inversely proportional to the Stokes wavelength, and also because the longer wavelength has a larger free-carrier absorption coefficient[68]. Currently, the minimum and maximum lasing wavelengths at 1325.5 and 1840.7 nm are limited by the pump wavelengths from the tunable lasers which have minimum and maximum wavelengths of 1240 and 1680 nm. Using similar pump powers in O-band and C-band wavelengths (Fig. 4a, b), we observed that the Raman lasing output powers with C-band pump wavelengths are two to three times larger than those with O-band pump wavelengths. The higher output power at the longer wavelength can be probably explained by the slightly higher $Q_i$ factors at the C band (Fig. 2f) resulting in lower Raman thresholds for the C-band wavelengths[68]. Overall, for this multimode concentric racetrack resonator, the Raman lasing outputs with C-band pumps are more efficient than those with O-band pumps.

## Discussion

We compare the performances of widely tunable Raman lasers on different platforms in Table 1. The tunable Raman lasers listed all

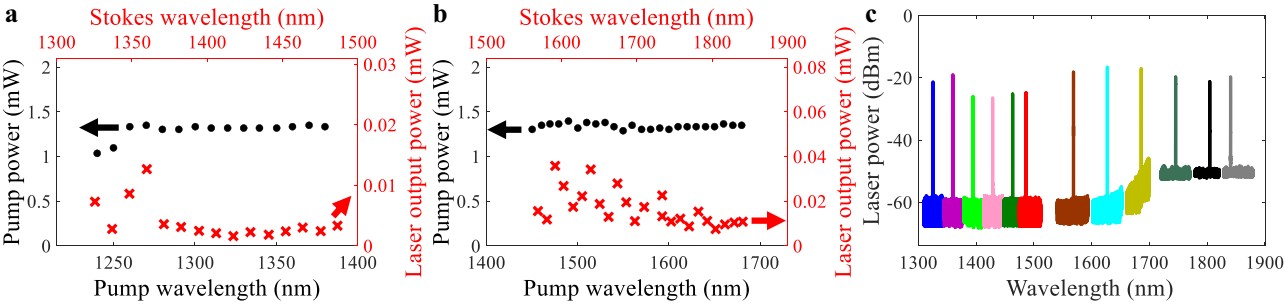

**Fig. 4 Stokes Raman lasing output characteristics. a, b** Pump powers used in the measurement at pump wavelengths with 10 nm spacings from 1240 to 1380 nm and from 1450 to 1680 nm. The corresponding Stokes laser output powers were observed in the Stokes wavelengths from 1325 to 1487 nm and from 1569 to 1841 nm. **c** Selective Stokes Raman laser output spectra at pump wavelengths of 1239.9, 1270.1, 1300.2, 1329.9, 1359.9, 1379.9, 1450.4, 1500.3, 1550, 1600.2, 1650.2, and 1680 nm.

**Table 1 Comparison of widely tunable Raman lasers.**

| Work | Platform | Lasing pump threshold (mW) | Lasing slope efficiency | Lasing tuning bandwidth (nm) |
|------|----------|---------------------------|------------------------|------------------------------|
| 33 | Diamond | 85 | 0.43% | 100 |
| 32 | Aluminum nitride | 8 | 3.6% | 90 |
| 30 | $As_2S_3$ glass | ≤3 | — | 53 |
| 31 | $Ge_{25}Sb_{10}S_{65}$ | 3.25 | 13.86% | 43 |
| 34 | Silicon | 15 | 26% | 83 |
| This work | Silicon | 0.4 ± 0.1 | (8.5 ± 1.5)% | 516 |

—: not reported in the paper.
Note: the lasing tuning bandwidth is for the first-order Stokes wave.

are tunable in discrete steps equal to the FSR of the resonator. For example, we recently reported silicon Raman lasing at O/S-band using a multimode racetrack resonator[68]. But that silicon Raman laser is not widely tunable, i.e., the pump wavelength cannot be continuously tuned across all FSRs because of the reduction in Q factors from coupling to higher-order modes at certain wavelengths. Further adding the temperature tuning on the chip can tune the resonances across one FSR and enable a continuous tuning range. That is, the Raman laser would not be limited to the tuning step of one FSR but can have arbitrary tuning wavelengths[31].

We demonstrated the use of multimode concentric racetrack resonators to achieve high Q factors across a broad wavelength range. Average loaded Qs of $1.4 \times 10^6$ were measured across a wavelength span of over 440 nm (from 1240 to 1680 nm) The multimode concentric racetrack resonators are composed of two multimode racetracks with widths of 1.5 and 3 μm. A strong mutual coupling can split the resonances to form two super-modes, which ensures at least one high-Q resonance exists in each FSR. Based on the resonator, a widely tunable Raman laser with an average tuning step of 0.25 nm (equaling one FSR of the resonator) is realized, and we demonstrated the tunability across a range of 516 nm with the Stokes output wavelengths covering the wavelength range from 1325 to 1841 nm (limited by the tuning range of the pump lasers). An average threshold pump power of (0.4 ± 0.1) mW and slope efficiency of (8.5 ± 1.5)% at 25 V reverse bias were measured experimentally.

The mechanism for maintaining broadband high Q factors is not to suppress the higher-order modes such as by using appropriately designed bends[42,47,48,50], but to use one of the two non-degenerate high-Q resonances of the multimode concentric racetracks to mitigate the mode coupling of the other high-Q resonance with the higher-order mode. The outer and inner resonators are composed of multimode waveguides with normal dispersion. The pulley directional coupler can be easily engineered to have comparable coupling coefficients over a wide

wavelength range. Thus, the multimode concentric resonators employed with the broadband pulley directional couplers can enable broadband high-Q resonances over several hundreds of nanometers wavelength ranges. This new approach does not suppress the higher-order modes but rather mitigates the reduction in Q factor within one FSR by providing an additional resonance near the mode coupling wavelengths. Even though the multimode resonator has normal dispersion, the presence of the mode and mutual coupling can modify the dispersion for the initialization of modulation instability which can be used for comb generation[53]. The multimode concentric resonators may therefore find applications beyond that demonstrated in the widely tunable Raman laser and may be of interest for use in wideband comb generation[65,69,70], or broadband integrated optical parametric oscillators[71].

Further integrating the on-chip Raman laser cavity with other passive structures and electronic circuits is promising for enabling wavelength conversion in silicon photonics and providing light sources that can extend to 2 μm or beyond for fully integrated systems on a single chip[72]. The broadband high-Q resonator may also find potential applications in efficient second-harmonic generation[73]. The longest lasing wavelength presented in this paper (1841 nm) was limited by the pump laser source and in principle. We can further extend the design for cascaded pumping to longer wavelengths. Generally, since the multimode concentric resonators address the problem of mode coupling with higher-order modes by having at least one of the two non-degenerate resonances to maintain a high Q factor in every FSR, any applications requiring a widely tunable wavelength range with low input power, broadband output, or individual high-Q resonances with large frequency detuning can find this approach useful.

## Methods

**Simulation**. We used commercial software (Lumerical solutions) to calculate the transmission of the pulley broadband directional coupler used in the calculation of

$Q_c$ factors (see Fig. 1b), and the eigenstates of supermodes in the concentric resonators (see Fig. 1c). The numerical simulation of the mode splitting (see Fig. 1e, f) is based on coupled-mode equations (Eqs. (2a–2c) and Eqs. (4a–4g)) to calculate the steady-state solution using commercial software, MATLAB.

**Measurement.** For the Raman lasing measurement, pump light from a tunable laser was coupled into the quasi-transverse-electric mode of the bus waveguide via the edge couplers and lensed fibers. An optical spectrum analyzer recorded the lasing spectrum output from the bus waveguide, and a power meter measured the Stokes output power. A source meter was used to apply the reverse bias voltages and measure the photocurrent. When the pump laser is tuned into resonance, there is a thermal shift in the resonance produced by the increased intracavity power, which shifts the resonance to longer wavelengths. Monitoring the photocurrent enables fine-tuning of the pump laser to a slightly longer wavelength and keeps it in the resonance despite the thermal shift of the resonator[18].

**Reporting summary.** Further information on research design is available in the Nature Research Reporting Summary linked to this article.

## Data availability

The data that support the findings of this study are included in the paper and its Supplementary Information. Other datasets are available from the corresponding author upon reasonable request.

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

## Acknowledgements

This work was supported by Hong Kong Research Grants Council, General Research Fund (RGC, GRF) (14207021), and the Research Matching Grant Scheme (RGMS) from University Grants Council in project number 8601438. Y.Z. would like to thank the support from the Postdoctoral Hub-Innovation and Technology Fund (PH-ITF). We would like to thank Dr. Hongnan Xu for useful discussion and AMF for device fabrication. We thank Synopsys Inc. for making their product Optodesigner available for us to produce the design layout of the photonic circuit.

## Author contributions

Y.Z. and K.Z. conceived the idea, designed the devices, and carried out the simulations. X.Z. designed the broadband grating couplers used in device comparison during the manuscript revision. Y.Z. and K.Z. conducted experiments. Y.Z., K.Z., and H.K.T. discussed the results and the theoretical models and wrote the original manuscript and its revised version. H.K.T. obtained the grants and supervised the projects.

## Competing interests

The authors declare no competing interests.
