## [Peer Review File · Nature Communications]

REVIEWER COMMENTS

Reviewer #2 (Remarks to the Author):

The authors demonstrate Raman lasing on silicon concentric racetrack resonators. This is a continuation of their recent Raman lasing study reported in *Laser Photonics Rev.* 15, 2000336 (2021), more focusing on broadband tuning capability by applying the concentric racetrack resonator structure, which was previously studied in *Nat. Commun.* 8, 1-8 (2017). The main claim is that mode coupling in a concentric resonator can mitigate the Q reduction in the multimode resonator. However, this claim is unclear, and I was unable to relate the role/need of a concentric resonator in this claim. The authors even did not differentiate between coupling-Q and total-Q. The claim is mostly about near-critical coupling, and the authors' claim can be fairly stated that they achieved near-critical coupling in the spectral region of 1240-1680 nm. Compared to previous works [e.g., *Opt. Lett.* 45, 4939 (2020)], this does not appear to be broadband. The authors may argue that broadband coupling in multimode is challenging. However, I think it is a matter of optimizing pulley coupling, and the concentric resonator is not necessarily required for broadband coupling. More detailed quantitative analysis is required if the authors claim concentric resonator's unique contribution to broadband coupling. Moreover, even if the concentric resonator assists the broadband coupling in some way, I still couldn't uncover the scientific innovation required for *Nature Communications*. Thus, I recommend the authors consider submitting this work to other journals.

Other comments:

- In Fig.1 caption, w_{in} and w_{out} seems switched (w_{in} should be larger than w_{out}).

- According to the author, the design details for the pulley coupler are described in the Supplementary. The Supplementary, on the other hand, only presents the generic features of pulley coupling qualitatively, with no coupling-Q calculations. If authors want to claim their design for achieving broadband near-critical coupling, they need to do a quantitative analysis. With the current form, it sounds like they accidentally achieved broadband and naively associated it with the concentric resonator.

- Efficiency calibration: the authors took into account varying coupling losses per wavelength. However, Si exhibits high Kerr nonlinearity and strong two-photon absorption. Thus, its fiber-chip and waveguide-resonator coupling efficiencies vary dramatically with pump power. If the resonances are scanned with varied pump powers, the coupling-Qs will change dramatically. It is suggested that the authors take this into account to calibrate the efficiency appropriately.

Reviewer #3 (Remarks to the Author):

The submission describes a novel design for multimode waveguides in a parallel configuration which solves the problem associated with higher mode coupling. This permits the development of high Q-factor racetrack resonators in the SOI waveguide system with implications for a range of non-linear devices. The underlying physical development of the analysis is sound, as one might expect from this group which has a reputation for both innovation and mathematical rigour. My sense is that this is an interesting new structure which will be of interest to a wide readership. However, before consideration for publication I ask the authors to address the following:

The authors should describe more broadly the potential for integrated Raman lasers in the SOI system. To my knowledge, there have been no wide deployment of silicon Raman lasers; in contrast to integrated lasers which utilize bonding of III-V structures on SOI.

Can the authors make a detailed comparison between their Raman lasers and those reported previously (for example Boyraz, O. & Jalali, B. Demonstration of a silicon Raman laser, *Opt. Express* 12, 5269–5273 (2004); Rong, H. et al. A continuous-wave Raman silicon laser, *Nature* 433, 725–728 (2005); Rong, H. et al. Low-threshold continuous-wave Raman silicon laser, *Nature Photon.* 1, 232–237 (2007); Jalali, B. et al. Prospects for silicon mid-IR Raman lasers, *IEEE J. Sel. Top. Quant. Electron.* 12, 1618–1627 (2006)), perhaps in the form of a table. I understand that they have done this for tunable Raman lasers, but not for silicon Raman lasers.

Could the authors also make a comparison with the performance of other optically pumped silicon lasers such as that recently report (Khadijeh Mirabbas Kiani,* Henry C. Frankis, Cameron M. Naraine, Dawson B. Bonneville, Andrew P. Knights, and Jonathan D. B. Bradley, *Laser Photonics Reviews* 2021, 2100348).

The authors should describe in which other non-linear applications the underlying principle of their work might be employed; i.e. please make a general case for your work as is required from Nature Comms.

The authors quote silicon mode SOI waveguides as having a propagation loss of ~ 2 dB/cm. However, in recent years the loss associated with silicon mode waveguides has been reduced due to advances in

immersion lithography, etchless waveguide fabrication and cladding engineering (see for example, Griffith A, Cardenas J, Poitras C B and Lipson M 2012 High quality factor and high confinement silicon resonators using etchless process Opt. Express 20 21341–21345; Horikawa T, Shimura D and Mogami T 2016 Low-loss silicon wire waveguides for optical integrated circuits MRS Communications 6 9–15; Nezhad M P, Bondarenko O, Khajavikhan M, Simic A and Fainman Y Etch-free low loss silicon waveguides using hydrogen silsesquioxane oxidation masks Opt. Express 19 18827–18832) . The authors should review developments in this area and explain why their approach remains significantly improved.

For Editor & Reviewer #1

Comment 1:

The authors demonstrate Raman lasing on silicon concentric racetrack resonators. (a) This is a continuation of their recent Raman lasing study reported in Laser Photonics Rev. 15, 2000336 (2021), more focusing on broadband tuning capability by applying the concentric racetrack resonator structure, which was previously studied in Nat. 347 Commun. 8, 1-8 (2017). The main claim is that mode coupling in a concentric resonator can mitigate the Q reduction in the multimode resonator. However, this claim is unclear, and I was unable to relate the role/need of a concentric resonator in this claim. (b) The authors even did not differentiate between coupling-Q and total-Q. (c) The claim is mostly about near-critical coupling, and the authors' claim can be fairly stated that they achieved near-critical coupling in the spectral region of 1240-1680 nm. (d) Compared to previous works [e.g., Opt. Lett. 45, 4939 (2020)], this does not appear to be broadband. The authors may argue that broadband coupling in multimode is challenging. However, I think it is a matter of optimizing pulley coupling, and the concentric resonator is not necessarily required for broadband coupling. More detailed quantitative analysis is required if the authors claim concentric resonator's unique contribution to broadband coupling. (e) Moreover, even if the concentric resonator assists the broadband coupling in some way, I still couldn't uncover the scientific innovation required for Nature Communications. Thus, I recommend the authors consider submitting this work to other journals.

Reply: We thank the reviewer for carefully checking our paper and appreciate the constructive comments which help to improve our paper better.

For the first question, we separate it into five parts and elaborate on each part separately.

(a) This is a continuation of their recent Raman lasing study reported in Laser Photonics Rev. 15, 2000336 (2021), more focusing on broadband tuning capability by applying the concentric racetrack resonator structure, which was previously studied in Nat. 347 Commun. 8, 1-8 (2017). The main claim is that mode coupling in a concentric resonator can mitigate the Q reduction in the multimode resonator. However, this claim is unclear, and I was unable to relate the role/need of a concentric resonator in this claim.

Reply: We thank the reviewer for this question.

The loaded Q factor is a parameter that depends on the total loss of a resonant cavity, including coupling loss and propagation loss. A cavity with broadband high loaded Q factors is desired for nonlinear applications, potentially useful for enabling lower input powers by enabling high enhancement over a broadband range. Firstly, we would like to explain why mode coupling can be a problem on the loaded Q factors in a multimode single resonator. For a multimode single resonator, the influence of mode coupling on loaded Q factors can be explained by Equations (R1-R2) with modeling the multimode single resonator with only fundamental (TE₀) mode and higher-order mode¹. Parameters A_0 and A_1 , ω_0 and ω_1 , τ_0

and τ_1 , κ_0 and κ_1 are the amplitudes, resonant frequencies, photon lifetimes, coupling coefficients of the TE0 mode and higher-order mode, respectively. u is the mutual coupling coefficient between the two modes. S_{in} is the amplitude of the input pump wave.

$$\frac{\partial A_0}{\partial t} = (j\omega_0 - \frac{1}{\tau_0})A_0 - j\kappa_0 S_{in} - juA_1 \quad (R1)$$

$$\frac{\partial A_1}{\partial t} = (j\omega_1 - \frac{1}{\tau_1})A_1 - j\kappa_1 S_{in} - juA_0 \quad (R2)$$

Fig. R1 | Diagrammatizing of a multimode single resonator.

In a multimode single resonator like Fig. R1, by assuming the TE0 mode and higher-order mode with parameters listed in Table R1, we solved equations (R1-R2) with different mutual coupling coefficients and obtained the transmission spectra in Fig. R2. Here, two adjacent wavelengths instead of equal resonant wavelengths are selected to display the independent loaded Q factors of the two modes by setting the mutual coupling coefficient u to be 0. In reality, u is a fixed value for a given structure. When the two resonant frequencies approach each other, amplitudes of the two modes would increase, leading to the enhancement of the coupling term $-juA_0$ or $-juA_1$.

Table R1 | Parameters of the multimode single resonator.

Mode	Resonant wavelength (nm)	Propagation loss α (dB/cm)	Coupling ratio $ k ^2$	Coupling condition
TE0	1555.5	0.26	6.9×10^8	under coupled
higher-order mode	1555.54	2.2	3.6×10^7	under coupled

Fig. R2 | Simulated transmission spectra of the multimode single resonator with different mutual coupling coefficients u .

In the simulation, we fixed the two resonant frequencies and gradually increased u . With the obtained transmission spectra in Fig. R2, we calculated the loaded Q factors for different u and summarized them in Table R2. With the increase of u , the loaded Q factor for TE0 mode gradually reduces. Because the resonances of the TE0 mode and a higher-order mode get closer in the wavelength domain. The TE0 mode thus suffers from additional effective loss from the coupling to higher-order mode and thus its loaded Q factor is reduced in the wavelength region where the resonances of the TE0 mode and a higher-order mode approach each other. The loaded Q factor of the TE0 mode can reduce by about 50% compared with that without the mode coupling as summarized in Table R2.

Table R2 | Calculated loaded Q factors for different u corresponding to Fig. R2.

u	Loaded Q factor (TE0)	Loaded Q factor (higher-order mode)
0	1.0×10^6	2.9×10^5
2×10^{10}	0.69×10^6	3.4×10^5
3×10^{10}	0.61×10^6	3.6×10^5
4×10^{10}	0.57×10^6	3.8×10^5
5×10^{10}	0.54×10^6	3.9×10^5

$$S_{in}^1 = \hat{t}_m S_m - j\hat{\kappa}_0 A_0 - j\hat{\kappa}_1 A_1, z \in [0, L_c] \quad (R3)$$

$$A_0^1 = \hat{t}_0 A_0 - j\hat{\kappa}_0 S_{in}, z \in [0, L_c] \quad (R4)$$

$$A_1^1 = \hat{t}_1 A_1 - j\hat{\kappa}_1 S_m, z \in [0, L_c] \quad (R5)$$

$$A_0^2 = \hat{t}_u A_0^1 - j\hat{\kappa}_u A_1^1 \quad (R6)$$

$$A_1^2 = \hat{t}_u A_1^1 - j\hat{\kappa}_u A_0^1 \quad (R7)$$

$$A_0 = A_0^2 e^{-(j\beta_0 + \frac{\alpha_0}{2})(L-L_c)}, z \in [L_c, L] \quad (R8)$$

$$A_1 = A_1^2 e^{-(j\beta_1 + \frac{\alpha_1}{2})(L-L_c)}, z \in [L_c, L] \quad (R9)$$

The performances of the two modes in the multimode single resonator can also be described by equations (R3-R9) under the assumption that the mode coupling [equations (R6-R7)] occurs immediately after the interaction between the two modes with bus waveguide [equations (R3-R5)]. Then, the two modes propagate along the resonator with phase change and loss [equations (R8-R9)]. $\hat{t}_0/\hat{\kappa}_0$, $\hat{t}_1/\hat{\kappa}_1$, $\hat{t}_u/\hat{\kappa}_u$ are amplitude transmission/coupling coefficients between the TE0 mode and bus waveguide, the higher-order mode and bus waveguide, the TE0 mode and higher-order mode, respectively. \hat{t}_{in} is the amplitude of the transmission coefficient of the bus waveguide and set to be $\sqrt{1 - \hat{\kappa}_0^2 - \hat{\kappa}_1^2}$. α and β are the propagation loss and constant. L_c is the coupling length between the bus waveguide and resonator, while L is the roundtrip length of the resonator.

Fig. R3 | Simulated and experimental transmission spectra of the multimode single resonator with two modes. Transmission spectra from 1553 nm to 1558 nm were obtained by the **a**, simulation and **b**, experiment.

Fig. R4 | Experimental transmission spectrum from 1522 nm to 1527 nm of the multimode single resonator with two modes.

By solving equations (R3-R9), we obtained the transmission shown in Fig. R3a. We can see that the mode coupling occurs at about 1556 nm. The loaded Q factor was calculated to be 4.9×10^5 compared to the loaded Q factor of 1.0×10^6 at 1553 nm which is away from the mode coupling. There is a 51% reduction in the loaded Q factor at the mode coupling region. The experimental transmission spectrum in Fig. R3b agreed well with the theoretical prediction in Fig. R3a. Some highly affected TE₀ mode resonances by the mode coupling can occur as the example in Fig. R4, the loaded Q factor decreased as 2.1×10^5 at a wavelength of 1524.1 nm compared to that of 1.1×10^6 in the non-mode coupling region. There is an 81% reduction of the loaded Q factor and 10 times larger loss at this resonance.

Compared to the multimode single resonator, the multimode concentric resonator in Fig. R5 displays a unique transmission property. Because the inner resonator does not directly interact with the bus waveguide. The TE₀ mode in the inner resonator (In_{TE0}) would not couple directly to the bus waveguide. The coupling only occurs between the TE₀ mode in the outer resonator (Out_{TE0}) and the In_{TE0} . When the Out_{TE0} and In_{TE0} couple to each other, they share the propagation loss. Considering the Out_{TE0} and In_{TE0}

with parameters listed in Table R3, we solved the equations (R10-R12)¹ and obtained the transmission spectra of the multimode concentric resonator with different mutual coupling coefficients u as shown in Fig. R6.

Fig. R5 | Diagrammatizing of a multimode concentric resonator.

$$\frac{dA_{out}}{dt} = (j\omega_{out} - \frac{1}{\tau_{out}})A_{out} - j\kappa S_{in} - juA_{in}, \quad (R10)$$

$$\frac{dA_{in}}{dt} = (j\omega_{in} - \frac{1}{\tau_{in}})A_{in} - juA_{out}, \quad (R11)$$

$$S_{out} = e^{-j\phi} (S_{in} - j\kappa^* A_{in}), \quad (R12)$$

Table R3 | Parameters of the multimode concentric resonator.

mode	Resonant wavelength (nm)	Propagation loss (dB/cm)	α	Coupling ratio $ \kappa ^2$	Coupling condition
In _{TE0}	1329.5	0.2		2%	Near critical
Out _{TE0}	1329.502	0.4		N.A.	N.A.

Fig. R6 | Simulated transmission spectra of the multimode concentric resonator with different mutual coupling coefficients u .

We calculated the loaded Q factors and summarized them in Table R4. We can see that the loaded Q factor of the Out_{TE0} slightly enhances as u increases. It can be understood that the Out_{TE0} with higher propagation loss couples to the In_{TE0} with lower propagation loss. The In_{TE0} shares the propagation loss of the Out_{TE0}. Thus, the loaded Q factor of the Out_{TE0} increases. In addition, also for the lower propagation loss of the inner resonator, when the mutual coupling happens, the In_{TE0} can present a higher loaded Q

factor than that of the Out_{TE0} without mutual coupling. In other words, two resonances with higher loaded Q factors than that of Out_{TE0} without mutual coupling can appear in one FSR as shown in Table R4. We found that both the Out_{TE0} and In_{TE0} can keep loaded Q factors over 10^6 after the interaction. That is, even though the mode coupling from the higher-order mode can highly decrease the loaded Q factor at TE0 mode, as present in the multimode single resonator, the existence of the two high-Q resonances at TE0 mode in the multimode concentric resonator can keep at least one high-Q resonance at TE0 mode in each free spectral range (FSR).

Table R4 | Calculated loaded Q factors for different u corresponding to Fig. R6.

u	Loaded Q factor (Out_{TE0})	Loaded Q factor (In_{TE0})
0	9.63×10^5	N.A.
1×10^9	1.07×10^6	2.41×10^6
2×10^9	1.2×10^6	2.11×10^6
3×10^9	1.29×10^6	1.93×10^6
4×10^9	1.34×10^6	1.83×10^6

$$S_{in}^1 = \hat{t}_{bus,out} S_{in} - j\hat{\kappa}_{bus,out} A_{out}, \quad z \in [0, L_c] \quad (\text{R13})$$

$$A_{out}^1 = -j\hat{\kappa}_{bus,out} S_{in} + \hat{t}_{out} A_{out} - j\hat{\kappa}_{out,in} A_{in}, \quad z \in [0, L_c] \quad (\text{R14})$$

$$A_{in}^1 = -j\hat{\kappa}_{out,in} A_{out} + \hat{t}_{out,in} A_{in}, \quad z \in [0, L_c] \quad (\text{R15})$$

$$\frac{\partial A_{out}^1}{\partial z} = (-j\beta_{out} - \frac{\alpha_{out}}{2}) A_{out}^1 - ju A_{in}^1, \quad z \in [L_c, L_c + L_{mc}] \quad (\text{R16})$$

$$\frac{\partial A_{in}^1}{\partial z} = (-j\beta_{in} - \frac{\alpha_{in}}{2}) A_{in}^1 - ju A_{out}^1, \quad z \in [L_c, L_c + L_{mc}] \quad (\text{R17})$$

$$A_{out} = A_{out}^2 e^{-(j\beta_{out} + \frac{\alpha_{out}}{2})(L_{out} - L_{mc} - L_c)}, \quad z \in [L_c + L_{mc}, L_{out}] \quad (\text{R18})$$

$$A_{in} = A_{in}^2 e^{-(j\beta_{in} + \frac{\alpha_{in}}{2})(L_{in} - L_{mc} - L_c)}, \quad z \in [L_c + L_{mc}, L_{in}] \quad (\text{R19})$$

Besides the above analyses in the time domain, the dynamics of concentric resonators can also be explained in the spatial domain, as indicated in equations (R13-R19)³. $\widehat{t}_{bus,out}/\widehat{\kappa}_{bus,out}$ and $\widehat{t}_{bus,in}/\widehat{\kappa}_{out,in}$ are amplitude transmission/coupling coefficients between the bus waveguide and outer resonator as well as the outer resonator and inner resonator. \widehat{t}_{out} is the amplitude transmission coefficient of the outer resonator and set to be $\sqrt{1 - \widehat{\kappa}_{bus,out}^2 - \widehat{\kappa}_{out,in}^2}$. u is the mutual coupling coefficient. α and β are the propagation loss and constant. L_c is the coupling length between the bus waveguide and resonator. L_{mc} is the mutual coupling length. Equations (R13-R15) depict electric field amplitudes variation of the bus waveguide, the outer and inner resonators in the coupling region, $[0, L_c]$. Then, we separate the remaining

process as the mutual coupling [equations (R16-R17)] and the pure propagation [equations (R18-R19)], since the roundtrip lengths of inner and outer resonators (L_{in} and L_{out}) are different. In simulation, we set $L_c = 0$ and $L_{mc} = L_{in}$ for simplification.

Fig. R7 | Simulated and experimental transmission spectra of the multimode concentric resonator. Transmission spectra from 1480 nm to 1481.5 nm were obtained by the **a**, simulation and **b**, experiment where the triangle marks the higher-order mode.

From the simulated transmission in Fig. R7a, both of the split two TE₀ resonances from the mutual coupling between the Out_{TE₀} and In_{TE₀} have loaded Q factors over 10^6 . The experimental spectrum in Fig. R7b was in good accordance with the simulation. From Fig. R7b, the higher-order mode marked with a triangle slightly couples to the split right TE₀ resonance, while the split left TE₀ resonance is barely affected. The above models and experiments well demonstrate how the multimode concentric resonator can mitigate the reduction in loaded Q factor due to the influence of higher-order modes on the TE₀ modes by introducing two high-Q resonance modes within one FSR induced by the interaction between the Out_{TE₀} and In_{TE₀}.

Fig. R8 | Microscope image of the multimode concentric resonator (up) and multimode single resonator (down). The zoomed-in image shows the clear views of two waveguides of the concentric resonator while one waveguide for the single resonator.

To further achieve broadband high Q factors in a multimode concentric resonator, a pulley directional coupler is useful. Because coupling coefficients with comparable values from 1200 nm to 1700 nm can be obtained by carefully engineering the pulley directional coupler. We then fabricated the multimode

concentric resonators with the designed pulley directional coupler and compared them with the multimode single resonators without additional inner racetracks in Fig. R8. The two types of resonators were fabricated in the same chip to avoid fabrication errors for clear comparison.

Fig. R9 | Experimental transmission spectrum of the multimode single resonator.

We first experimentally characterized the transmission spectra of the two resonators. Even employing the broadband pulley directional coupler, we found that the multimode single resonator still suffers from the effects from the higher-order modes periodically in Fig. R9. Because the higher-order modes spectral resonances can become nearly coincident with the fundamental modes periodically.

Fig. R10 | Experimental transmission spectra of the multimode single resonator (left) and multimode concentric resonator (right). The triangle marks represent the higher-order modes.

We magnify the affected resonances of the multimode single resonator at the mode coupling region in Fig. R10a. As the higher-order mode (triangle mark) gets close to the TE₀ mode, more loss is coupled into the TE₀ mode which results in a high decrease in the extinction ratio and the loaded Q factor. However, for the same wavelength range of the multimode concentric resonator in Fig. R10b, due to the split two high-Q TE₀ resonances at 1591 nm, even though the higher-order mode slightly catches the right TE₀ resonance, the left TE₀ resonance is barely affected. The principle of concentric resonators mitigating loaded Q factor reduction can be stated as follow. When a TE₀ mode couples to a higher-order mode with a much larger propagation loss in a multimode single resonator, the loaded Q factor would be reduced

seriously. However, in a multimode concentric resonator, when Out_{TE0} couples to In_{TE0} with slightly smaller propagation loss, both consequently split resonances present higher loaded Q factors comparable to that of the main resonant mode without mutual coupling. Therefore, by adding an inner resonator, we add an additional high-Q resonance within one FSR when independent resonant wavelengths of Out_{TE0} and In_{TE0} approach each other. If one mode encounters higher-order mode and suffers from a serious reduction in loaded Q factor, there is still another mode propagating mainly in another resonator with displaying high loaded Q factor.

In detail, when a higher-order mode slightly couples to a TE0 mode, they share the coupling loss and propagation loss of each other. However, the propagation loss of the higher-order mode can be up to 8 times larger than that of the TE0 mode. Therefore, in the mode coupling region, the effective propagation loss of the TE0 mode would increase. Conventionally, in a multimode single resonator, the higher-order mode is unwanted and its coupling from the bus waveguide should be suppressed. The small coupling ratio of the higher-order mode can lower the effective coupling ratio of the TE0 mode when they couple to each other and share coupling ratios. Therefore, if the reduced coupling loss of the TE0 mode is smaller than its increased propagation loss, the total loss of TE0 mode still increases, leading to the reduced loaded Q factor. TE0 mode always presents the reduced loaded Q factor in the mode coupling region for near-critical and under coupling cases. On the contrary, when the reduced coupling loss of the TE0 mode is larger than its increased propagation loss, the enhancement of the loaded Q factor occurs. We observed such a phenomenon in experiments only when TE0 mode is seriously over coupled with a low extinction ratio of only 6 dB. For nonlinear application, the cavity usually would not be so seriously over coupled where the loaded Q factor is quite small. Even though it can get a higher loaded Q factor from the mode coupling, most resonances are still with low Q factors. Therefore, we didn't consider it here. In our designs, the main resonant mode which is away from the mode coupling region presents slightly over coupling with a large extinction ratio of 15 dB under a low pump power from the laser. We classify the slightly over-coupling with extinction ratio over 10 dB into the near-critical coupling of which the loaded Q factors are still high at the order of 10^6 .

Fig. R11 | Loaded Q factors comparison between the multimode single and multimode concentric resonators. The ratio of loaded Q factor at mode coupling regions ($Q_{L,MC}$) over the nearby average loaded Q factors but slightly away from mode coupling regions ($Q_{L,NMC}$) from 1200 nm to 1700 nm in the

multimode single and the multimode concentric resonators at **a**, under coupling and **b**, near-critical coupling conditions.

By varying the pulley directional couplers, we designed and fabricated two types of multimode single and multimode concentric resonators with under coupling and near-critical coupling in Fig. R11. We compared the ratio of the loaded Q factors at all the mode coupling regions over the nearby average loaded Q factors but slightly away from the mode coupling regions. At both under-coupling and near-critical coupling conditions, all the mode regions in the multimode concentric resonators exhibit higher loaded Q factor ratios than those in the multimode single resonators, indicating smaller reductions of the loaded Q factors in the mode coupling regions.

As above mentioned, because of the mutual coupling from the inner resonator with lower propagation loss, two resonances with higher loaded Q factors than of Out_{TE0} without mutual coupling can occur. If the higher-order mode affects one mode, there remains another mode presenting a high loaded Q factor. That is why Q_{L_MC}/Q_{L_NMC} can be larger than 1 in some mode coupling regions, as indicated in Fig. R11. We fabricated two pairs of the above devices in the same chip to consider the fabrication errors. All the devices can improve 97% loaded Q factors at all the mode coupling regions. The remaining 3% regions are the places where the mode coupling slightly affects the loaded Q factors. That is, the multimode concentric resonators can highly alleviate the reductions on loaded Q factors in the mode coupling regions. Besides, from Fig. R11, the near-critical coupling resonators can have less than 36% mode coupling regions compared to the under-coupling resonators. Generally, the multimode concentric resonators work well to mitigate the decrease of the loaded Q factors at the mode coupling regions and achieve the broadband high Q factors over a wide wavelength range.

To further validate our results experimentally, we fabricated two additional multimode concentric resonators. We kept the gaps between the bus waveguide and the outer racetracks as 400 nm. But we varied the gaps between the inner and outer racetracks as 500 nm and 300 nm in the two devices. Compared to the original device with a gap of 400 nm, the number of the mode coupling regions has increased by 45% and 19%. It may imply that the gap between the inner and outer racetracks is an important aspect to engineer for fewer mode coupling regions. Another two multimode concentric resonators with changing the width of the inner racetrack as 1.5 μm (the same width of the outer racetrack) and 3.25 μm were also fabricated in the same chip. The number of the mode coupling regions has increased by 15% and 6%. Very large reductions of the loaded Qs in the multimode concentric resonator with 1.5- μm -width inner racetrack were found in three of the mode coupling regions. A wide inner racetrack can reduce the loss, but it can also induce more higher-order modes. Thus, careful engineering of the widths of the inner racetrack is also critical. To reduce the higher-order modes in the multimode concentric resonator, other approaches, like single-mode bends, Euler bends, and Bezier bends are useful. For resonators, maintaining a high Q factor is important in some nonlinear devices because the high Q can directly enable low-input power devices. The enhancement factor, denoting the build-up of light intensity

in the cavity, is defined as the light intensity coupled into the ring over the light intensity in the bus waveguide³:

$$M = \frac{I_p(0)}{I_{in}} = \frac{1-t^2}{(1-at)^2} \quad (\text{R21})$$

t is the transmission coefficient, a is the dimensionless loss coefficient.

Fig. R12 | Enhancement factor comparison between the multimode single and multimode concentric resonators. The ratio of the enhancement factor at mode coupling regions (M_{MC}) over the nearby average enhancement factor but slightly away from mode coupling regions (M_{NMC}) from 1200 nm to 1700 nm in the multimode single and the multimode concentric resonators at **a**, under coupling and **b**, near-critical coupling conditions.

We calculated the ratios of the enhancement factors in the mode coupling region over the enhancement factors out of the mode coupling region in Fig. R12. Generally, the multimode concentric resonators exhibit smaller variation in enhancement factor than these of the multimode single resonators, since M_{MC}/M_{NMC} is closer to one in Fig. R12 for the multimode concentric resonators. For the values of the enhancement factors, the multimode concentric resonators also manifest larger values compared to the multimode single resonators at the mode coupling regions.

As an example of the benefit of using the multimode concentric resonator, we demonstrated its use in an integrated widely tunable Raman laser, where the enhancement factor directly affects the required input pump power to achieve Raman lasing threshold P_{th} as follows⁴.

$$P_{th} = \frac{\alpha_s A_{eff}}{g_r MT} \quad (\text{R22})$$

α_s is the linear loss coefficient for the Stokes, A_{eff} is the effective area, g_r is the Raman gain coefficient and T is the transmission factor (output power over input power). In the mode coupling region, we separately chose one resonance of the multimode single resonator and one resonance of the multimode concentric resonator to compare their Raman lasing threshold powers. They have corresponding enhancement factors as 50.8 and 15.3. Using equation R21, we can calculate the Raman lasing threshold power as 0.3 mW and 1.3 mW. Fig. R13 shows the Raman lasing threshold powers were measured as 0.5 mW and 1.5 mW, in

consonance with the theoretical predictions. Besides, due to the highly decreased Q factor in the mode coupling region of the multimode single resonator, the Stokes output power is largely reduced by 500 times smaller at the similar pump powers. If using the highly affected resonance with the largely decreased Q factor in Fig. R4, the theoretical threshold power increases to be 94 mW, which is 200 times larger than that in the multimode concentric resonator.

Fig. R13 | Raman lasing threshold power characteristics at the mode coupling regions. Stokes Raman laser output powers as a function of pump powers in **a**, multimode single resonator and **b**, multimode concentric resonator applied with 25 V reverse biases.

In summary, through adding theoretical models and conducting measurements, all the results comprehensively proved that the multimode concentric resonator can well mitigate the mode coupling from the higher-order mode to keep at least one high-Q resonance at TE₀ mode in each FSR over a wide wavelength range compared to the highly decreased Q factors at the mode coupling regions in the multimode single resonator. Using the multimode concentric resonator as the application of a widely tunable Raman laser, the multimode concentric resonator can keep low threshold power compared to the over 200 times larger threshold power in the multimode single resonator.

1. Zhang Z, Dainese M, Wosinski L, Qiu M. Resonance-splitting and enhanced notch depth in SOI ring resonators with mutual mode coupling. *Opt. Express* **16**, 4621-4630 (2008).
2. Okamoto K. *Fundamentals of optical waveguides*. Elsevier, (2021).
3. Chen C-L. *Foundations for guided-wave optics*. John Wiley & Sons, (2006).
4. Zhang Y, Zhong K, Tsang HK. Raman lasing in multimode silicon racetrack resonators. *Laser Photonics Rev.* **15**, 2000336 (2021).

In the revision, we have added the above models and experimental results of the multimode single resonator and multimode concentric resonator in the sections “S1. Mode coupling in a multimode single resonator”, “S2. Mode coupling in a multimode concentric resonator” and “S3. Broadband high-Q multimode concentric resonator” of the Supplementary Information. More descriptions of the mode coupling are added in the last paragraph of section “INTRODUCTION” on page 7 and the revised Fig. 1e and corresponding descriptions on page 6 for explaining the mutual coupling in the concentric resonator of the revised paper as follows:

“In principle, when the resonant frequencies of the fundamental modes of inner and outer resonators

approach each other, mutual coupling enables mode splitting to form symmetric and anti-symmetric modes. One mode propagates mainly in the outer resonator, while the other is mainly in the inner resonator. Because the inner resonator is designed to have a larger width than the outer resonator, the mode in the inner resonator has a smaller propagation loss. In this scenario, the mutual coupling between the two resonators slightly reduced the effective propagation loss of the mode in the outer resonator, while that of mode coupling can slightly increase the effective loss in the inner resonator. Since both modes have low propagation losses, after the mutual coupling, two comparable high-Q resonances could generate within one FSR. Even if one high-Q resonance couples with higher-order mode leading to seriously reduced loaded Q factor, there remains the other resonance to maintain one high-Q resonance in the same FSR.”

Fig. 1 | The concentric racetrack resonator design. e, Simulated transmission spectra around 1550 nm with different coupling coefficients u between the inner and outer racetracks. f, Simulated transmission spectrum of the multimode concentric resonator from 1480 nm to 1481.5 nm.

“Because the inner resonator does not directly interact with the bus waveguide, In_{TE0} does not couple directly to the bus waveguide. The coupling only occurs between Out_{TE0} and In_{TE0} . When Out_{TE0} and In_{TE0} couple to each other, they share the propagation loss. We assume the propagation losses of the outer and inner racetracks are 0.4 dB/cm and 0.2 dB/cm⁴³. Considering the Out_{TE0} and In_{TE0} with parameters listed in Table S3 (see section 2 of the Supplementary Information), we solved the above equations and obtained the transmission spectra of the multimode concentric resonator with different mutual coupling coefficients u as shown in Fig. 1e. The mode splitting generates anti-symmetric and symmetric modes. The two modes produced from the mutual coupling propagate along the optical paths between the two racetracks.”

Besides the theoretical model in the time domain to explain the mutual coupling in the concentric resonator, we added the model in the spatial domain and more analyses on page 6 and 7. And the corresponding figures about the model and compares to the experimental result of the revised paper in Fig. 1f and Fig. 2e as follows:

$$S_{in}^1 = \hat{t}_{bus,out} S_{in} - j\hat{\kappa}_{bus,out} A_{out}, z \in [0, L_c] \quad (4a)$$

$$A_{out}^1 = -j\hat{\kappa}_{bus,out} S_{in} + \hat{t}_{out} A_{out} - j\hat{\kappa}_{out,in} A_{in}, z \in [0, L_c] \quad (4b)$$

$$A_{in}^1 = -j\hat{\kappa}_{out,in} A_{out} + \hat{t}_{out,in} A_{in}, z \in [0, L_c] \quad (4c)$$

$$\frac{\partial A_{out}^1}{\partial z} = (-j\beta_{out} - \frac{\alpha_{out}}{2}) A_{out}^1 - ju A_{in}^1, z \in [L_c, L_c + L_{mc}] \quad (4d)$$

$$\frac{\partial A_{in}^1}{\partial z} = (-j\beta_{in} - \frac{\alpha_{in}}{2}) A_{in}^1 - ju A_{out}^1, z \in [L_c, L_c + L_{mc}] \quad (4e)$$

$$A_{out} = A_{out}^2 e^{-(j\beta_{out} + \frac{\alpha_{out}}{2})(L_{out} - L_{mc} - L_c)}, z \in [L_c + L_{mc}, L_{out}] \quad (4f)$$

$$A_{in} = A_{in}^2 e^{-(j\beta_{in} + \frac{\alpha_{in}}{2})(L_{in} - L_{mc} - L_c)}, z \in [L_c + L_{mc}, L_{in}] \quad (4g)$$

Besides the above analyses in the time domain, the dynamics of concentric resonators can also be explained in the spatial domain⁵⁸, in equations (4a-4g). $\hat{t}_{bus,out}/\hat{\kappa}_{bus,out}$ and $\hat{t}_{bus,in}/\hat{\kappa}_{out,in}$ are the amplitude transmission/coupling coefficients between bus waveguide and outer resonator, and between the outer resonator and inner resonator. \hat{t}_{out} is the amplitude transmission coefficient of the outer resonator and set to be $\sqrt{1 - \hat{\kappa}_{bus,out}^2 - \hat{\kappa}_{out,in}^2}$. u is the mutual coupling coefficient. α and β are the propagation loss and constant. L_c is the coupling length between the bus waveguide and resonator. L_{mc} is the mutual coupling length. Equations (4a-c) depict electric field amplitudes variation of the bus waveguide, the outer and inner resonators in the coupling region, $[0, L_c]$. We separate the mutual coupling [equations (4d-4e)] and the pure propagation [equations (4f-4g)], since the roundtrip lengths of inner and outer resonators (L_{in} and L_{out}) are different. In simulation, we set $L_c = 0$ and $L_{mc} = L_{in}$ for simplification. By solving the above equations, we obtained the simulated transmission in Fig. 1f, both of the split two TE0 resonances from the mutual coupling between Out_{TE0} and In_{TE0} have loaded Q factors over 10^6 .

Fig. 2 | Transmission characteristics of the concentric resonator. d, e. Selected transmission spectra

region around 1329 nm and 1480 nm where the inverted triangle marks the higher-order mode.

The experimental spectrum in Fig. 2e agreed well with the simulation in Fig. 1f. From Fig. 2e, the higher-order mode (inverted triangle) slightly couples to the split right TE0 resonance, while the split left TE0 resonance is barely affected. The above models and experiments well prove that the multimode concentric resonator can help to mitigate the reduction in loaded Q factor due to the influence of higher-order modes on the TE0 modes by introducing two high-Q resonance modes within one FSR induced by the interaction between the Out_{TE0} and In_{TE0}. On the contrary, in a multimode single resonator, the highly decreased loaded Q factors induced by the mode coupling from higher-order modes can be up reduced by 81%. The corresponding theoretical models and experimental results are included in Section 1, Section 2 and Section 3 of the Supplementary Information.

(b) The authors even did not differentiate between coupling-Q and total-Q.

Reply: We thank the reviewer for raising this point.

The Q factor is used to characterize the loss of energy in a resonator. Generally, we regard the Q factor as the loaded Q (Q_L) factor when the resonator couples with the bus waveguide. Two parts of coupling Q (Q_c) factor and intrinsic quality (Q_i) factor contribute to the Q_L factor. The Q_c factor depends on the coupling coefficient κ between the resonator and the bus waveguide while the Q_i factor relates to intrinsic cavity power loss coefficient α . The Q_c , Q_i , and Q_L factors can be expressed as⁵⁴⁻⁵⁶

$$Q_c = \omega \frac{n_g^R}{c_0} \frac{2\pi R}{|\kappa|^2} \quad (\text{R23})$$

$$Q_i = \frac{2\pi n_g}{\alpha \lambda} = \frac{2Q_L}{1 \pm \sqrt{T_0}} \quad (\text{R24})$$

$$Q_L = \frac{\lambda}{\text{FWHM}} \quad (\text{R25})$$

$$\frac{1}{Q_L} = \frac{1}{Q_i} + \frac{1}{Q_c} \quad (\text{R26})$$

where ω and λ are the resonant frequency and wavelength; n_g is the group index; R is the radius of the resonator; FWHM is the full width at half maximum of the resonance; T_0 is the normalized transmission at the resonance. Equation R24 takes the – sign under the over-coupled condition and + sign under the under-coupled condition.

To characterize the coupling condition between the bus waveguide and outer resonator in the multimode concentric resonator, we selected a wavelength section away from mode coupling and swept transmission spectra with different input powers. We chose a resonant wavelength around 1331.4 nm as shown in Fig. R14. With the increased powers, the extinction ratio firstly increases at input powers from 0.1 mW to 1.6 mW (over coupled) and then decreases at input powers from 1.6 mW to 2 mW (under coupled). It could

be understood that the nonlinear losses induced by two-photon absorption and free-carrier absorption tend to increase as the power goes up, changing the coupling conditions from over coupling to critical coupling and finally to under coupling. The redshift of resonant wavelength with increasing input powers is due to the Kerr effect and thermo-optic effect¹.

Fig. R14 | Experimental Transmission spectra with different input powers.

Using the above equations and the conclusion of the over coupling condition at low power, we obtain the Q_c and Q_L factors from the measured transmission data in Fig. R15. Both Q_c and Q_L factors have high values over 10^6 from the wavelength range of 1260 nm to 1650 nm. Generally, a high-Q resonator means the large build-up of the light intensity in the cavity. It is a key to achieving low-power devices. Broadband high-Q resonators are useful for applications with widely tunable ranges at low powers.

Fig. R15 | Measured loaded quality (Q_L) and coupling Q (Q_c) factors from 1260 nm to 1650 nm.

1. Zhang L, Fei Y, Cao Y, Lei X, Chen S. Experimental observations of thermo-optical bistability and self-pulsation in silicon microring resonators. *JOSA B* **31**, 201-206 (2014).

To better differentiate the Q_c and Q_L factors, we have added the plot of Q_c and Q_L factors against wavelengths and some references which include the equations for the calculations of the Q_c and Q_L factors for the readers' references in the revised paper. We also include the equations to calculate the Q_c and Q_L factors in the "S2. Resonator characteristics" of the Supplementary Information. The corresponding changes have been underlined and marked with red color on page 9 of the revised paper as follows:

“The coupling Q (Q_c) and loaded Q (Q_L) factors of the TE₀ modes at resonant wavelengths from 1260 nm to 1360 nm and 1450 nm to 1650 nm in each FSR are shown in Fig. 2f. Both Q_c and Q_L factors exhibit relatively flat variations from 1260 nm to 1650 nm, indicating the well-designed pulley directional coupler for broadband resonator-waveguide coupling⁵⁹⁻⁶¹. Even though we design the resonator to have broadband near-critical coupling, the fabricated resonator seems to be slightly over coupled from 1260 nm to 1650 nm.

59. Moille G, Li Q, Briles TC, Yu S-P, Drake T, Lu X, *et al.* Broadband resonator-waveguide coupling for efficient extraction of octave-spanning microcombs. *Opt. Lett.* **44**, 4737-4740 (2019).

60. Moille G, Perez EF, Stone JR, Rao A, Lu X, Rahman TS, *et al.* Ultra-broadband Kerr microcomb through soliton spectral translation. *Nat. Commun.* **12**, 1-9 (2021).

61. Xuan Y, Liu Y, Varghese LT, Metcalf AJ, Xue X, Wang P-H, *et al.* High-Q silicon nitride microresonators exhibiting low-power frequency comb initiation. *Optica* **3**, 1171-1180 (2016).

Fig. 2 | Transmission characteristics of the concentric resonator. f, Coupling Q (Q_c), and loaded Q (Q_L) factors of the TE₀ modes at resonant wavelengths from 1260 – 1360 nm and 1450 – 1650 nm (the absent wavelength range covering 1360 – 1450 nm is limited by the tunable lasers).

S6. Resonator characteristics

For an all-pass resonator, the loaded quality (Q_L) factor can be expressed as^{S3}:

$$Q_L = \frac{\lambda}{\text{FWHM}} = \frac{\pi n_g L \sqrt{ra}}{\lambda(1-ra)}, \quad (\text{S12})$$

where FWHM is the full width at half maximum of the resonance; n_g is the group velocity; $a = \exp(-\alpha L/2)$ is the amplitude transmission and α is the power attenuation coefficient; L is the roundtrip length of the resonator; λ is the resonant wavelength; r is the self-coupling coefficient of the directional coupler. The intrinsic quality (Q_i) factor and coupling Q (Q_c) factor can be calculated by^{S4-6}:

$$Q_i = \frac{2\pi n_g}{\alpha \lambda} = \frac{2Q_L}{1 \pm \sqrt{T_0}}, \quad (\text{S13})$$

$$Q_c = \omega \frac{n_g}{c_0} \frac{2\pi R}{|\kappa|^2} \quad (\text{S14})$$

where ω is the resonant frequency; R is the radius of the resonator; T_0 is the normalized transmission at the resonance. Equation S13 takes + sign for the over-coupled regime while – sign for the under-coupled regime.

S3. Bogaerts W, De Heyn P, Van Vaerenbergh T, De Vos K, Kumar Selvaraja S, Claes T, et al. Silicon microring resonators. *Laser Photonics Rev.* 6, 47-73 (2012).

S4. Xuan Y, Liu Y, Varghese LT, Metcalf AJ, Xue X, Wang P-H, et al. High-Q silicon nitride microresonators exhibiting low-power frequency comb initiation. *Optica* 3, 1171-1180 (2016).

S5. Moille G, Li Q, Briles TC, Yu S-P, Drake T, Lu X, et al. Broadband resonator-waveguide coupling for efficient extraction of octave-spanning microcombs. *Opt. Lett.* 44, 4737-4740 (2019).

S6. Moille G, Perez EF, Stone JR, Rao A, Lu X, Rahman TS, et al. Ultra-broadband Kerr microcomb through soliton spectral translation. *Nat. Commun.* 12, 1-9 (2021).

(c) The claim is mostly about near-critical coupling, and the authors' claim can be fairly stated that they achieved near-critical coupling in the spectral region of 1240-1680 nm.

Reply: We thank the reviewer for the comment.

To present the near-critical coupling in the spectral region of 1240-1680 nm, the condition of $Q_c \approx Q_i$ is required to be satisfied. In the simulation, we designed the pulley direction coupler to have comparable coupling ratios of 2.82% from 1200 nm to 1700 nm. Assuming that the propagation loss is 0.45 dB/cm, the resonator is expected to have broadband critical coupling. Experimentally, same as the above answer. As shown in Fig. R14, with the increased powers, the extinction ratio firstly increases at input powers from 0.1 mW to 1.6 mW (over coupled) and then decreases at input powers from 1.6 mW to 2 mW (under coupled). At low power, the extinction ratio of the resonance is still larger than 10 dB so that the resonator is slightly over coupled.

Fig. R14 | Transmission spectra with different input powers.

We derived the measured Q_i and Q_c factors in Fig. R16. The Q_i factors are slightly larger than the Q_c factors, of which the difference is within 3 times. Both Q_i and Q_c factors are larger than 10^6 . Even though we designed the resonator to be near-critical coupling from 1240 nm to 1680 nm, the fabrication tolerances may result in the resonator being slightly over coupled from 1240 nm to 1680 nm. Therefore,

our statement of the advantage of the multimode concentric resonator is that it can maintain high Q factors over a broadband range.

Fig. R16 | Measured intrinsic quality (Q_i) and coupling Q (Q_c) factors.

To clarify the comment of having broadband near-critical coupling of the resonator, we added one sentence underlined and marked with red color in the first paragraph on page 9 of the revised paper as follows

“Even though we designed the resonator to have broadband near-critical coupling, the fabricated resonator seems to be slightly over coupled from 1260 nm to 1650 nm.”

(d) Compared to previous works [e.g., Opt. Lett. 45, 4939 (2020)], this does not appear to be broadband. The authors may argue that broadband coupling in multimode is challenging. However, I think it is a matter of optimizing pulley coupling, and the concentric resonator is not necessarily required for broadband coupling. More detailed quantitative analysis is required if the authors claim concentric resonator’s unique contribution to broadband coupling.

Reply: We thank the reviewer for this comment.

For a single-mode waveguide-based single resonator, a pulley directional coupler can be used to realize broadband coupling for TE₀ mode and thus enable broadband nonlinear application⁵⁹. However, a pulley coupler cannot ensure high Q factors for a resonator with a multimode waveguide over a wide wavelength range. Mode coupling with a higher-order mode having larger propagation loss can introduce large losses than that to the TE₀ mode. Even though we can design a pulley directional coupler with comparable coupling ratios over a wide wavelength range, we still found the corresponding multimode single resonator suffers from the mode coupling losses periodically. By adding an inner resonator in a multimode concentric resonator, it can alleviate the highly decreased Q factors at the mode coupling regions by inducing an additional high-Q TE₀ resonance.

We fabricated the two types of resonators in the same chip for comparison as shown in Fig. R8. The detailed comparisons are included in the answer to question a, which will prove the multimode concentric resonator's contribution for broadband high Q factors. We believe that if no other means are implemented,

such as single-mode bends, Euler bends, and Bezier bends, mode coupling could still destroy broadband high Q factors even with a broadband pulley directional coupler.

Fig. R8 | Microscope image of the multimode concentric resonator (up) and multimode single resonator (down).

59. Moille G, Li Q, Briles TC, Yu S-P, Drake T, Lu X, *et al.* Broadband resonator-waveguide coupling for efficient extraction of octave-spanning microcombs. *Opt. Lett.* **44**, 4737-4740 (2019).

(e) Moreover, even if the concentric resonator assists the broadband coupling in some way, I still couldn't uncover the scientific innovation required for Nature Communications. Thus, I recommend the authors consider submitting this work to other journals.

Reply: We thank the reviewer for this comment, which showed that we did not explain well the novelty in our work. In response to the question on our scientific innovation, we developed a new model that showed, for the first time, how a multimode concentric resonator with a broadband directional coupler can enable the multimode waveguide resonator to maintain high Q over a very broad wavelength region 1240 nm to 1680 nm. This model solves the common problem in multimode single resonators in which some, in certain wavelength regions, the resonator cannot maintain high Q factors because of the unavoidable mode coupling between the higher-order mode and fundamental mode which will decrease the Q factor of the fundamental mode. The decreased Q factors can be five times lower than the highest Q factor in the same resonator, as the same from the previous report in a multimode Si₃N₄ resonator⁴⁹. According to the theory, we fabricated several devices, and all verified the designs from the theoretical model well. Using the broadband high-Q resonator, we experimentally demonstrate the most widely tunable Raman laser until now covering the tunability across a range of 516 nm from 1325 nm to 1841 nm (limited by the tuning range of the pump lasers). The Raman laser has an average low lasing threshold power of (0.4 ± 0.1) mW with considering both the mode coupling and non-mode coupling regions.

To emphasize the scientific innovation of our work novelty, we have added a statement on novelty by making the following changes, which have been underlined and marked with red color on page 3 of the revised paper as follows:

“The multimode concentric resonator well ameliorates the problem of localized regions of the frequency spectrum having large decreases in Q factors arising from the degeneracy of resonant frequencies at the mode coupling regions in multimode resonators. The concentric resonator allows high Q factors to be maintained in every single FSR of over a broad optical bandwidth.”

Comment 2:

Other comments:

- In Fig.1 caption, w_{in} and w_{out} seems switched (w_{in} should be larger than w_{out}).

Reply: Thanks for the reviewer's carefully checking. We have corrected the w_{in} and w_{out} in Fig.1 caption as w_{in} : 3 μm ; w_{out} : 1.5 μm . The corresponding changes have been underlined and marked with red color in the Fig.1 caption of the revised paper as follows:

“Fig. 1 | The concentric racetrack resonator design. a, Structure of the concentric racetrack resonator along with a p-i-n junction to remove the free carriers. **b,** Schematic from the top view with geometric parameters. w_0 : 0.88 μm ; w_{in} : 3 μm ; w_{out} : 1.5 μm ; gap1: 0.4 μm ; gap2: 0.4 μm ; R_{out} : 200 μm ; L_s : 690 μm .”

Comment 3:

According to the author, the design details for the pulley coupler are described in the Supplementary. The Supplementary, on the other hand, only presents the generic features of pulley coupling qualitatively, with no coupling-Q calculations. If authors want to claim their design for achieving broadband near-critical coupling, they need to do a quantitative analysis. With the current form, it sounds like they accidentally achieved broadband and naively associated it with the concentric resonator.

Reply: Thanks for the reviewer's suggestion.

We have added the coupling-Q calculations and more designs of the multimode concentric resonators with near-critical and under coupling with different pulley directional couplers. All the quantitative analyses indicate that the multimode concentric resonators with pulley directional couplers can well ameliorate the decreased Q factors at the mode coupling regions to achieve broadband high Q factors. The details can be found in the answer to question a.

Comment 4:

Efficiency calibration: the authors took into account varying coupling losses per wavelength. However, Si exhibits high Kerr nonlinearity and strong two-photon absorption. Thus, its fiber-chip and waveguide-resonator coupling efficiencies vary dramatically with pump power. If the resonances are scanned with varied pump powers, the coupling-Qs will change dramatically. It is suggested that the authors take this into account to calibrate the efficiency appropriately.

Reply: Thanks for the reviewer's suggestion.

As the reviewer mentioned, due to the large two-photon absorption of the silicon, the fiber-chip and waveguide resonator coupling efficiency vary dramatically with pump power. Here, the lasing slope efficiency is the slope in the plot of the Stokes output power after being coupled out to the bus waveguide

as a function of the pump power coupled into the bus waveguide before coupling into the resonator. Even though the resonator has several tens of enhancement factors, the pump power coupled into the bus waveguide was 1.3 mW in the measurements of the Raman lasing. And the Raman lasing experiments are under the reverse bias of 25 V. The free carriers are well depleted and can be ignored.

However, since the decreased Q factors in the mode coupling regions, we added the Raman lasing threshold measurement at the mode coupling region in Fig. R13. The measurements in Fig. R13 show the Raman lasing threshold powers and slope efficiency were measured as 0.5 mW and 7%. Together with the lasing threshold power and slope efficiency measured out of the mode coupling region as 0.3 mW and 10%, the average lasing threshold power and slope efficiency were measured to be (0.4 ± 0.1) mW and (8.5 ± 1.5) %.

Fig. R13 | Raman lasing threshold power characteristics at the mode coupling region. Stokes Raman laser output powers as a function of pump powers in the multimode concentric resonator applied with a 25 V reverse bias.

To express the definition of the lasing slope efficiency more clearly, we added the descriptions of the Stokes output power and the pump power used to derive the lasing slope efficiency in the measurement of Raman lasing threshold power. The corresponding changes have been underlined and marked with red color in the last paragraph on page 9 of the revised paper as follows:

“We first measure the Stokes output power (after being coupled out to the bus waveguide) as a function of pump power (in the bus waveguide before coupling into the resonator) to get the lasing threshold power at the pump wavelength of 1550 nm shown in Fig. 3.”

We also revised the lasing threshold power and the slope efficiency at the mode coupling region to have average values in the first paragraph on page 10 of the revised paper as follows:

“Including the lasing threshold power of 0.5 mW and slope efficiency of 7% at the mode coupling region (see Fig. S11, Supplementary Information), the average lasing threshold power and slope efficiency of this resonator were measured as (0.4 ± 0.1) mW and (8.5 ± 1.5) %. The experimental results agreed well with the theoretical predictions in Section 4 of the Supplementary Information.”

For Editor & Reviewer #2

Comment 1:

The submission describes a novel design for multimode waveguides in a parallel configuration which solves the problem associated with higher mode coupling. This permits the development of high Q-factor racetrack resonators in the SOI waveguide system with implications for a range of non-linear devices. The underlying physical development of the analysis is sound, as one might expect from this group which has a reputation for both innovation and mathematical rigour. My sense is that this is an interesting new structure which will be of interest to a wide readership. However, before consideration for publication I ask the authors to address the following:

The authors should describe more broadly the potential for integrated Raman lasers in the SOI system. To my knowledge, there have been no wide deployment of silicon Raman lasers; in contrast to integrated lasers which utilize bonding of III-V structures on SOI.

Reply: We thank the reviewer for carefully checking our paper and appreciate the constructive comments which help to improve our paper better.

As the reviewer mentioned, compared to integrated lasers which utilize bonding of III-V structures on SOI, integrated Raman lasers in the SOI platform have not been fully developed for commercial use. The advantage of III-V lasers on the SOI platform is their low-cost, energy-efficient, and wafer-scale photonic integrated circuits. The III-V lasers are electrically driven while the integrated Raman lasers are optically pumped. Besides, the Raman laser doesn't need dispersion engineering to satisfy the phase-matching condition. On-chip Raman lasers usually employ cavity designs with tightly compact footprints. Further embedded the integrated Raman laser with finely tuned thermal heaters can enable the laser with the tuning resolution of 0.1 pm, which is comparable with that of the commercial semiconductor laser.

Recent progress using hybrid integration of an InP distributed feedback (DFB) laser with silicon photonics and complementary metal-oxide-semiconductor electronic circuits demonstrated the first complete on-chip solution for microwave photonics system⁶⁴. Instead of using the InP DFB laser as the light source, utilizing the on-chip Raman laser as the light source can be further integrated with other passive structures to achieve a fully on-chip silicon photonics system in one single silicon chip.

We added more discussion about the potential for integrated Raman lasers in the SOI system for readers' reference in the last paragraph on page 12 in the revised paper. The corresponding changes have been underlined and marked with red color as follows,

“Further integrating the on-chip Raman laser cavity with other passive structures and electronic circuits is promising for enabling wavelength conversion in silicon photonics and providing light sources that can extend to 2 μm or beyond for fully integrated systems on a single chip⁶⁴.”

64.Tao Y, Yang F, Tao Z, Chang L, Shu H, Jin M, *et al.* Fully on-chip microwave photonics system. *arXiv preprint arXiv:2202.11495* (2022).

Comment 2:

Can the authors make a detailed comparison between their Raman lasers and those reported previously (for example Boyraz, O. & Jalali, B. Demonstration of a silicon Raman laser, *Opt. Express* 12, 5269–5273 (2004); Rong, H. et al. A continuous-wave Raman silicon laser, *Nature* 433, 725–728 (2005); Rong, H. et al. Low-threshold continuous-wave Raman silicon laser, *Nature Photon.* 1, 232–237 (2007); Jalali, B. et al. Prospects for silicon mid-IR Raman lasers, *IEEE J. Sel. Top. Quant. Electron.* 12, 1618–1627 (2006)), perhaps in the form of a table. I understand that they have done this for tunable Raman lasers, but not for silicon Raman lasers. Could the authors also make a comparison with the performance of other optically pumped silicon lasers such as that recently report (Khadijeh Miarabbas Kiani,* Henry C. Frankis, Cameron M. Naraine, Dawson B. Bonneville, Andrew P. Knights, and Jonathan D. B. Bradley, *Laser Photonics Reviews* 2021, 2100348).

Reply: Thanks for the reviewer’s suggestion.

The above-mentioned works are the milestones of the silicon Raman lasers. We have cited all in the revised paper as below.

“SRS is mediated by a coherent phonon population and does not require phase matching between the pump and Stokes waves. Raman lasers are therefore useful to produce output wavelengths at longer wavelengths than available from the pump laser⁷⁻²³.”

5. Rong H, Xu S, Kuo Y-H, Sih V, Cohen O, Raday O, et al. Low-threshold continuous-wave Raman silicon laser. *Nat. Photonics* 1, 232 (2007).

8. Boyraz O, Jalali B. Demonstration of a silicon Raman laser. *Opt. Express* 12, 5269-5273 (2004).

21. Rong H, Jones R, Liu A, Cohen O, Hak D, Fang A, et al. A continuous-wave Raman silicon laser. *Nature* 433, 725 (2005).

22. Jalali B, Raghunathan V, Shori R, Fathpour S, Dimitropoulos D, Stafsudd O. Prospects for silicon mid-IR Raman lasers. *IEEE J. Sel. Top. Quantum Electron.* 12, 1618-1627 (2006).

23. Miarabbas Kiani K, Frankis HC, Naraine CM, Bonneville DB, Knights AP, Bradley JD. Lasing in a hybrid rare - earth silicon microdisk. *Laser Photonics Rev.* 16, 2100348 (2022).

The comparison table is added:

Table S5 | Comparison between integrated silicon Raman lasers.

work	Type	Lasing threshold (W)	Lasing slope efficiency	Bias (V)	Lasing wavelength band	Cascaded Raman order
s ⁹	Pulse	9	8.5%	0	C band	-
S ¹⁰	cw	0.182	4.3%	25	C band	-
S ¹¹	cw	0.02	28%	25	C band	-
S ¹²	cw	-	-	-	Mid-infrared	3
S ¹³	cw	0.016	4.2%	0	1.9 μm	-
This work	cw	(0.4 ± 0.1) × 10 ⁻³	(8.5 ± 1.5)%	25	C band	-

-: not present in the paper.

We have added the summarized table and the corresponding changes have been underlined and marked with the red color in the “S3. Comparison on silicon Raman lasers” of the revised Supplementary Information as follows,

S7. Comparison on silicon Raman lasers

In Table S5, we summarize the progress on integrated silicon Raman lasers from the pulsed Raman laser to continuous-wave Raman laser^{S9, 10}. Attempts on developing low-threshold Raman laser and extending the lasing wavelength band from C-band to the mid-infrared region have also been included^{S11, 12}. Further approach on hybrid silicon with rare-earth can reduce the Raman lasing threshold power and extend the lasing wavelength to 1.9 μm , which is promising for mid-infrared applications^{S13}.

Table S5 | Comparison between integrated silicon Raman lasers.

work	Type	Lasing threshold (W)	Lasing slope efficiency	Bias (V)	Lasing wavelength band	Cascaded Raman order
s9	Pulse	9	8.5%	0	C band	=
S10	cw	0.182	4.3%	25	C band	=
S11	cw	0.02	28%	25	C band	=
S12	cw	=	=	=	Mid-infrared	3
S13	cw	0.016	4.2%	0	1.9 μm	=
This work	cw	$(0.4 \pm 0.1) \times 10^{-3}$	$(8.5 \pm 1.5)\%$	25	C band	=

:- not present in the paper.

S9. Boyraz O, Jalali B. Demonstration of a silicon Raman laser. *Opt. Express* **12**, 5269-5273 (2004).

S10. Rong H, Jones R, Liu A, Cohen O, Hak D, Fang A, *et al.* A continuous-wave Raman silicon laser. *Nature* **433**, 725 (2005).

S11. Rong H, Xu S, Kuo Y-H, Sih V, Cohen O, Raday O, *et al.* Low-threshold continuous-wave Raman silicon laser. *Nat. Photonics* **1**, 232 (2007).

S12. Jalali B, Raghunathan V, Shori R, Fathpour S, Dimitropoulos D, Stafsudd O. Prospects for silicon mid-IR Raman lasers. *IEEE J. Sel. Top. Quantum Electron.* **12**, 1618-1627 (2006).

S13. Miarabbas Kiani K, Frankis HC, Naraine CM, Bonneville DB, Knights AP, Bradley JD. Lasing in a hybrid rare - earth silicon microdisk. *Laser Photonics Rev.* **16**, 2100348 (2022).

Comment 3:

The authors should describe in which other non-linear applications the underlying principle of their work might be employed; i.e. please make a general case for your work as is required from Nature Comms.

Reply: Thanks for the reviewer’s suggestion.

Here, the multimode concentric resonator with a broadband directional coupler achieved broadband high Q factors from 1240 nm to 1680 nm. Other nonlinear applications like broadband frequency combs and

second-harmonic generation are promising. The low-power dark-pulsed frequency combs are desirable in this multimode concentric resonator of which the mode coupling would help to comb initialization⁵⁶. For the second-harmonic generation, applying electrical components to break the inversion symmetry of the silicon and satisfying the quasi-phase-matching can be used to achieve efficient broadband second-harmonic signal with low pump power⁶⁰. Generally, since the multimode concentric resonators have ameliorated the common problem of mode coupling from higher-order mode to periodically affect the Q factors and thus present broadband high Q factors, the applications requiring widely tunable wavelength range with low input power, broadband output or individual high-Q resonances with large frequency detuning can be feasible.

We have other two potential nonlinear applications using this broadband high-Q multimode concentric resonator in the last paragraph on page 12 of the revised paper as follows. The correspondingly related works were also cited in the revised paper for readers' reference.

“The broadband high-Q resonator may also find potential applications in broadband frequency combs⁶⁰, and efficient second-harmonic generation⁶⁵. The longest lasing wavelength presented in this paper (1841 nm) was limited by the pump laser source and in principle, we can further extend the design for cascaded pumping to longer wavelengths. Generally, since the multimode concentric resonators address the problem of mode coupling with higher-order modes by having at least one of the two non-degenerate resonances maintaining a high Q factor in every FSR, any applications requiring widely tunable wavelength range with low input power, broadband output, or individual high-Q resonances with large frequency detuning can find this approach useful.”

60. Moille G, Perez EF, Stone JR, Rao A, Lu X, Rahman TS, *et al.* Ultra-broadband Kerr microcomb through soliton spectral translation. *Nat. Commun.* **12**, 1-9 (2021).

65. Singh N, Raval M, Ruocco A, Watts MR. Broadband 200-nm second-harmonic generation in silicon in the telecom band. *Light Sci. Appl.* **9**, 1-7 (2020).

Comment 4:

The authors quote silicon mode SOI waveguides as having a propagation loss of ~2dB/cm. However, in recent years the loss associated with silicon mode waveguides has been reduced due to advances in immersion lithography, etchless waveguide fabrication and cladding engineering (see for example, Griffith A, Cardenas J, Poitras C B and Lipson M 2012 High quality factor and high confinement silicon resonators using etchless process Opt. Express 20 21341–21345; Horikawa T, Shimura D and Mogami T 2016 Low-loss silicon wire waveguides for optical integrated circuits MRS Communications 6 9–15; Nezhad M P, Bondarenko O, Khajavikhan M, Simic A and Fainman Y Etch-free low loss silicon waveguides using hydrogen silsesquioxane oxidation masks Opt. Express 19 18827–18832) . The authors should review developments in this area and explain why their approach remains significantly improved.

Reply: Thanks for the reviewer’s suggestion.

To achieve a high-Q resonator, waveguides with low propagation loss are required. The 450-nm SOI waveguides fabricated by the commercial foundries typically have standard propagation losses around 2 dB/cm and lead to loaded Q factor only in the order of 10^5 .

As the reviewer mentioned, recent reports have impressively realized less than 1 dB/cm propagation loss for silicon nano waveguide. To achieve low propagation loss, they improved the fabrication methods including the etchless thermal oxidation^{38, 39}, high-resolution ArF immersion lithography⁴⁰, and post-etching roughness removal processes⁴¹. We summarized the detailed information on the development in this area in Table R5.

Table R5 | Development of low-loss silicon nano waveguide.

Year	Width (nm)	Height (nm)	Slab (nm)	Wavelength (nm)	Loss (dB/cm)	Method
2011 ³⁸	600	125	40	1599.69	0.35	Thermal oxidation with HSQ mask
2012 ³⁹	500	300	\	1550	0.9	Thermal oxidation with LPCVD grown SiN mask
2016 ⁴⁰	440 (320)	220	\	1550 (1310)	0.4 (1.28)	ArF immersion lithography
2020 ⁴¹	400	300	\	1550 (1310)	0.7 (1.1)	Si ₃ N ₄ mask and H ₂ thermal annealing

38. Nezhad MP, Bondarenko O, Khajavikhan M, Simic A, Fainman Y. Etch-free low loss silicon waveguides using hydrogen silsesquioxane oxidation masks. *Opt. Express* **19**, 18827-18832 (2011).

39. Griffith A, Cardenas J, Poitras CB, Lipson M. High quality factor and high confinement silicon resonators using etchless process. *Opt. Express* **20**, 21341-21345 (2012).

40. Horikawa T, Shimura D, Mogami T. Low-loss silicon wire waveguides for optical integrated circuits. *MRS Commun.* **6**, 9-15 (2016).

41. Wilmart Q, Brisson S, Hartmann J-M, Myko A, Ribaud K, Petit-Etienne C, *et al.* A complete Si photonics platform embedding ultra-low loss waveguides for O-and C-band. *J. Lightwave Technol.* **39**, 532-538 (2020).

In our case, since we used the commercial foundry for fabrication, we chose a multimode waveguide to obtain low propagation loss and further to achieve a high-Q resonator. The wider multimode waveguides offer smaller propagation losses for the fundamental mode because of the reduced modal overlap with the sidewall roughness. Therefore, our fabricated multimode waveguide racetrack resonators can achieve loaded Q factors over 10^6 . Using the reviewer mentioned above methods to fabricate the nano waveguide based resonator can also highly increase the Q factor with the reduced propagation losses.

We clarified the statement that the propagation loss of 2 dB/cm in the single-mode waveguide is fabricated by a commercial multi-project wafer foundry and added the description about the development of the reduced propagation loss in the nano waveguide and cited these papers in the last paragraph on page 2 of the revised paper as follows:

“However, high-index-contrast single-mode silicon waveguide-based resonators fabricated by commercial multi-project wafer foundries suffer from high propagation losses of typically about 2 dB/cm mainly produced by the scattering losses associated with sidewall roughness and surface defects. The highest Q factors of these resonators³⁷ are usually limited to around 10⁵. Further fabrication processes can be used to reduce the losses of the single-mode waveguides³⁸⁻⁴¹.”

The above-compared table was added in the section "S9. Development of low-loss silicon nano waveguide" of the supplementary information for readers' reference. The corresponding changes have been underlined and marked with red color as follows:

“S9. Development of low-loss silicon nano waveguide

The 450-nm SOI waveguides fabricated by a commercial multi-project wafer foundry have standard propagation losses around 2 dB/cm. The wider multimode waveguides offer smaller propagation losses for the fundamental mode because of the reduced modal overlap with the sidewall roughness. Therefore, our fabricated multimode waveguide racetrack resonators can achieve loaded Q factors over 10⁶. Using the reviewer mentioned above methods to fabricate the nano waveguide based resonator can also highly increase the Q factor with the reduced propagation losses. Multimode waveguide racetrack resonators have routinely achieved loaded Q factors of well over 10⁶. Recent progress on nano waveguides can achieve propagation loss as low as 0.4 dB/cm by optimizing the fabrication such as the etchless thermal oxidation^{S14, 15}, high-resolution ArF immersion lithography^{S16}, and post-etching roughness removal processes^{S17}. They are summarized in Table S6. We used the commercial foundry for fabrication so that we use the multimode waveguide to obtain low loss and further achieve a high-Q resonator.”

Table S6 | Development of low-loss silicon nano waveguide.

Year	Width (nm)	Height (nm)	Slab (nm)	Wavelength (nm)	Loss (dB/cm)	Method
2011^{S14}	600	125	40	1599.69	0.35	Thermal oxidation with HSQ mask
2012^{S15}	500	300	\	1550	0.9	Thermal oxidation with LPCVD grown SiN mask
2016^{S16}	440 (320)	220	\	1550 (1310)	0.4 (1.28)	ArF immersion lithography
2020^{S17}	400	300	\	1550 (1310)	0.7 (1.1)	Si₃N₄ mask and H₂ thermal annealing

S14. Nezhad MP, Bondarenko O, Khajavikhan M, Simic A, Fainman Y. Etch-free low loss silicon waveguides using hydrogen silsesquioxane oxidation masks. *Optics Express* **19**, 18827-18832 (2011).

S15. Griffith A, Cardenas J, Poitras CB, Lipson M. High quality factor and high confinement silicon resonators using etchless process. *Optics express* **20**, 21341-21345 (2012).

S16. Horikawa T, Shimura D, Mogami T. Low-loss silicon wire waveguides for optical integrated circuits. *MRS Communications* **6**, 9-15 (2016).

S17. Wilmart Q, Brisson S, Hartmann J-M, Myko A, Ribaud K, Petit-Etienne C, *et al.* A Complete Si Photonics Platform Embedding Ultra-Low Loss Waveguides for O-and C-Band. *Journal of Lightwave Technology* **39**, 532-538 (2020).

Thanks for your valuable time. We greatly appreciate the comments which help to make the paper better.

Yours sincerely,

Dr. Yaojing Zhang, Ms. Keyi Zhong, Mr. Xuetong Zhou, and Dr. Hon Ki Tsang

Center for Advanced Research in Photonics

Department of Electronic Engineering

The Chinese University of Hong Kong, Shatin, Hong Kong

Email: hktsang@ee.cuhk.edu.hk

REVIEWER COMMENTS

Reviewer #1 (Remarks to the Author):

The authors clearly pointed out that the role of the concentric resonator in this work is to help avoid the coupling of the fundamental (that has a high-Q) with a higher-order mode, which reduces the loaded-Q caused by a higher loss. However, I still question its significance for the following reasons:

1) Basically, the concentric resonator introduces two fundamental TE₀ modes, one at the inner and the other at the outer. From Fig.1c, the outer resonator supports a single mode and the inner resonator supports two modes, TE₀ and TE₁. In other words, the concentric resonator here is more likely the combination of one single-mode resonator and one two-mode resonator, whose intrinsic-Qs are expected to be lower than a multimode resonator with a wider width. (Note that the intrinsic-Qs should be compared as it relates to the nonlinear efficiency.) If the widths of the inner and outer resonators of the concentric resonator widen, the concentric resonator will exhibit a similar mode coupling to a higher-order mode. So, it cannot be claimed that the concentric resonator helps avoid coupling to a higher-order mode, rather the specific scheme presented here simply does not exhibit a higher-order mode (enough to see such effects).

2) The authors may claim there is a higher-order mode with the TE₁ mode. As the authors have shown, it causes a mode coupling at some spectral locations reducing the Q. The authors commented that there are two fundamental TE₀ modes, and even if TE₁ couples to one of the TE₀ modes, the other TE₀ can be utilized. As it said here, the coupling to a higher-order mode is there, and such use of the other TE₀ mode is practically limited as different mode groups show different modal properties like FSRs and dispersions.

3) Moreover, even if the presented scheme mitigates Q (at particular spectral regimes of mode coupling) by avoiding coupling to a higher-order mode in a limited case, its effect and scientific novelty are questionable. For example, a higher-order mode filtering, as demonstrated in *Opt. Lett.* 41, 452 (2016), sounds more effective and generally applicable ways to avoid the coupling to a higher-order mode that the authors mainly claim in this paper.

4) In addition, the authors presented temporal coupled modes for describing the mode couplings in spectral resonances. However, temporal coupled modes are an abstractive representation of the given phenomena plugging/fitting unknown parameters. It is an intuitive way to describe the couplings, but it does not allow ones to design or model the devices. Thus, I disagree with the authors' claim that they presented a new model to model/develop a resonator avoiding the coupling.

In my opinion, the presented work catches eyes at first glance and is well-studied work for a specific experiment. However, I disagree with the novelty and the significance that the authors claim and recommend this work be published in other journals.

Other comments:

- Instead of plotting the coupling fractions in Fig.1b, it would be better to plot in coupling-Q so that others can compare it with the coupling-Qs in Fig.2f.
- Fig.1f: y-axis label is recommended to be changed.

Reviewer #2 (Remarks to the Author):

The authors have provided detailed replies to all the points raised by both reviewers. I recommend this paper for publication.

Dear Editor and Reviewer,

Thank you again for reviewing our revised paper, “Broadband high-Q multimode silicon concentric racetrack resonators for widely tunable Raman lasers” (NCOMMS-21-42517A). The authors of this paper are Yaojing Zhang, Keyi Zhong, Xuotong Zhou, and Hon Ki Tsang. We would like to thank the first reviewer for his/her very insightful comments, which further improved the paper. In the resubmitted version of the paper, we have addressed all of the comments by point-by-point responses. Our responses to the comments and amendments made to the paper are listed in this letter, together with the reviewer’s original comments (*blue and italic*). All the changes are underlined and marked with red color in this letter and the revised paper. The corrections we have made are listed as follows:

Revisions/Explanation

For Editor & Reviewer #1

The authors clearly pointed out that the role of the concentric resonator in this work is to help avoid the coupling of the fundamental (that has a high-Q) with a higher-order mode, which reduces the loaded-Q caused by a higher loss. However, I still question its significance for the following reasons:

Comment 1:

1) (a) Basically, the concentric resonator introduces two fundamental TE₀ modes, one at the inner and the other at the outer. From Fig.1c, the outer resonator supports a single mode and the inner resonator supports two modes, TE₀ and TE₁. (b) In other words, the concentric resonator here is more likely the combination of one single-mode resonator and one two-mode resonator, whose intrinsic-Qs are expected to be lower than a multimode resonator with a wider width. (Note that the intrinsic-Qs should be compared as it relates to the nonlinear efficiency.) (c) If the widths of the inner and outer resonators of the concentric resonator widen, the concentric resonator will exhibit a similar mode coupling to a higher-order mode. So, it cannot be claimed that the concentric resonator helps avoid coupling to a higher-order mode, rather the specific scheme presented here simply does not exhibit a higher-order mode (enough to see such effects).

Reply: We sincerely appreciate the reviewer’s constructive comments and suggestions that help us to improve and present the manuscript more clearly.

For the first question, we separate it into three parts and elaborate on each part separately.

(a) For the confined modes in our multimode concentric resonator, we showed only the fundamental mode of the outer resonator in the manuscript which might have misled the reviewer to think that the outer resonator is single mode. Actually, the outer resonator waveguide width is 1.5 μm , and it can support three quasi-transverse-electric modes as the simulated mode profiles in Fig. R1. For the wavelength range that we consider from 1240 nm to 1680 nm, the Out_{TE₂} mode is well confined at the short wavelength and gradually becomes less confined at the longer wavelengths. The inner resonator has a width of 3 μm

which can support at least five modes as the below simulated mode profile. That is, both the inner and outer resonators are multimode and the whole concentric resonator is multimode.

Fig. R1 | Simulated mode propagates mainly in the outer racetrack, and modes are excited in the inner racetrack.

(b) We agree with the reviewer that the intrinsic Q (Q_i) factor is larger in a wider multimode resonator in which the propagation loss is smaller. Because the Q_i factor is inversely proportional to the propagation loss as the following equation for the calculation of the Q_i factor¹.

$$Q_i = \frac{2\pi n_g}{\alpha \lambda}, \quad (\text{R1})$$

n_g is the group velocity; α is the power attenuation coefficient; λ is the resonant wavelength.

Fig. R2 | Intrinsic Q (Q_i) factors of the TE0 modes at resonant wavelengths from 1200 – 1700 nm.

That is exactly the reason that we employ multimode waveguide widths for both outer and inner resonators to obtain low losses so that the whole multimode concentric resonator is low. The measured Q_i factors from 1200 nm to 1700 nm have values from 2.3×10^6 to 7.3×10^6 with an average value of 3.7×10^6 . in Fig. R2. The average propagation loss of the multimode concentric resonator is thus calculated to be 0.3 ± 0.15 dB/cm. As the reviewer mentioned, we used the wide multimode waveguide for low loss, but we do not make it wider than necessary for high Q factor because we also want high nonlinear efficiency in the application for the broadly tunable Raman silicon laser to get a low threshold power of (0.4 ± 0.1) mW and a slope efficiency of (8.5 ± 1.5) % at 25 V reverse bias.

(c) And also, as the reviewer mentioned that if the widths of the inner and outer resonators in the concentric resonator widen to be multimode, the common problem of mode coupling from a higher-order mode to the fundamental mode will occur. This is exactly the novelty of our work that we mitigate this mode coupling problem in the multimode resonator by adding a multimode inner resonator to the multimode resonator to construct a multimode concentric resonator. The functions of the inner resonators were well elucidated in our first revision by comparing the single multimode resonators and the multimode concentric resonators.

In summary, the multimode concentric resonator in our work indeed consists of two multimode outer and inner resonators. The outer resonator supports three modes, and the inner resonator supports five modes. Since both inner and outer resonators are consist of multimode waveguides, the average loss and intrinsic Q factor of the multimode concentric resonator at the wavelength range from 1200 nm to 1700 nm are as low as 0.3 ± 0.15 dB/cm and high as 3.7×10^6 . The multimode concentric resonators were well demonstrated to mitigate the mode coupling problems in single multimode resonators in our first revision.

1. Xuan Y, Liu Y, Varghese LT, Metcalf AJ, Xue X, Wang P-H, *et al.* High-Q silicon nitride microresonators exhibiting low-power frequency comb initiation. *Optica* **3**, 1171-1180 (2016).

To avoid misunderstanding and clarify that the multimode concentric resonator has multimode outer and multimode inner resonators, we added more descriptions of the structure on page 4 and page 5 in the revised manuscript as follows:

“Broadband high-Q multimode concentric racetrack resonator design. The resonator is comprised of two multimode concentric racetracks, which are coupled to each other, and the outer racetrack is coupled to a bus waveguide.”

“Both outer and inner racetrack resonators are multimode indicated by the existence of higher-order modes in the simulated mode profiles in Fig. S10 (see section 4 of the Supplementary Information).”

We added the intrinsic Q factors of our multimode concentric resonator at wavelengths from 1260 nm to 1650 nm in Fig. 2f and the corresponding description of the high intrinsic Q factors and the low loss on page 9 in the revised manuscript as follows:

Fig. 2 | Transmission characteristics of the concentric resonator. **a**, Microscope image of the concentric resonator with zoom-in direction coupler region in the inset. **b, c**, Transmission spectra with wavelength ranges from 1260 – 1360 nm and 1450 – 1650 nm. **d, e**, Selected transmission spectra around 1329 nm and 1480 nm where the inverted triangle marks the higher-order mode. **f**, Intrinsic Q (Q_i), coupling Q (Q_c), and loaded Q (Q_l) factors of the TE₀ modes at resonant wavelengths from 1260 – 1360 nm and 1450 – 1650 nm (the absent wavelength range covering 1360 – 1450 nm is limited by the tunable lasers).

“The measured results show that the multimode concentric resonator maintains high Q factors over the wavelength range from 1260 nm to 1650 nm, with average Q_l and Q_i factors of 1.4×10^6 and 3.7×10^6 . The average propagation loss of the resonator is calculated to be 0.3 ± 0.15 dB/cm.”

The simulated mode profiles in the multimode concentric resonator are detailed included in section 4 of the supplementary information as follows:

“S4. Modes in the multimode concentric resonator

Fig. S10 | Simulated mode propagates mainly in the outer racetrack, and modes are excited in the inner racetrack.

The simulated eigenstates of the supermodes in the concentric racetrack resonator are shown in Fig. S10. For the wavelength range from 1240 nm to 1680 nm, the TE₂ mode in the multimode outer resonator is strong at a short wavelength and gradually becomes less confined in the long wavelength. The inner resonator has a width of 3 μm which can support at least five modes as the below simulated mode profile. That is, the multimode concentric resonator is composed of multimode waveguides for both the outer and inner resonators.”

Comment 2:

2) The authors may claim there is a higher-order mode with the TE₁ mode. As the authors have shown, it causes a mode coupling at some spectral locations reducing the Q. The authors commented that there are two fundamental TE₀ modes, and even if TE₁ couples to one of the TE₀ modes, the other TE₀ can be utilized. As it said here, the coupling to a higher-order mode is there, and such use of the other TE₀ mode is practically limited as different mode groups show different modal properties like FSRs and dispersions.

Reply: We thank the reviewer for this question.

We agree with the reviewer that the existence of the coupling to a higher-order mode would modify the modal properties like FSRs and dispersions. It is common in normal multimode single resonators that the mode coupling to higher-order modes can perturb the FSRs and local dispersions. In the case of the multimode concentric resonator, there are two fundamental TE₀ modes (symmetric and anti-symmetric supermodes, which have split resonances), and even if destructive higher-order mode couples to one of the TE₀ modes, the other TE₀ can be utilized.

Fig. R3 | Measured FSRs and dispersions. Dispersion is described by parameter $D_{int} = \omega_{\mu} - \omega_0 - \mu D_1$, where ω_{μ} is the angular resonant frequency of mode order μ and D_1 is FSR in angular frequency.

We further measured the FSRs and dispersions of three different modes in the desired mutual coupling region with equal optical path lengths for the fundamental modes of inner and outer resonators, around 1330 nm. As displayed in Fig. R3, when mutual coupling occurs, symmetric and anti-symmetric TE₀ modes are formed, and their FSRs and dispersions would be changed due to interaction with each other. Even though both fundamental modes of inner and outer resonators have normal dispersion, the mutual coupling enables the anti-symmetric TE₀ mode to have anomalous dispersion which can be further employed for comb generation, consistent with the previous report in Nat. Commun. 8, 1-8 (2017). That is, the variations of modal properties in the concentric resonators have more possible applications, like comb generation apart from our demonstrated broadband Raman laser. Apart from the two TE₀ modes, the third mode is believed to be In-TE₁ for its intermediate magnitude of FSR and almost unchanged modal properties. The abrupt changes of FSR and dispersion experienced first by symmetric mode, then by In-TE₁ mode, and finally by anti-symmetric mode during the wavelength sweep from short to long wavelengths, are believed to result from the interaction with higher-order mode successively.

More discussion about the modified FSRs and dispersions are added on page 13 of the revised paper as follows:

“This new approach does not suppress the higher-order modes but rather mitigates the reduction in Q factor within one FSR by providing an additional resonance near the mode coupling wavelengths. Even though the multimode resonator has normal dispersion, the presence of the mode and mutual coupling can modify the dispersion for the initialization of modulation instability which can be used for comb generation⁵³. The multimode concentric resonators may therefore find applications beyond that

demonstrated in the widely tunable Raman laser and may be of interest for use in wideband comb generation⁶², or broadband integrated optical parametric oscillators⁶⁶.”

53. Kim S, Han K, Wang C, Jaramillo-Villegas JA, Xue X, Bao C, *et al.* Dispersion engineering and frequency comb generation in thin silicon nitride concentric microresonators. *Nat. Commun.* **8**, 1-8 (2017).

62. Moille G, Perez EF, Stone JR, Rao A, Lu X, Rahman TS, *et al.* Ultra-broadband Kerr microcomb through soliton spectral translation. *Nat. Commun.* **12**, 1-9 (2021).

66. Kuyken B, Liu X, Osgood RM, Baets R, Roelkens G, Green WM. A silicon-based widely tunable short-wave infrared optical parametric oscillator. *Opt. Express* **21**, 5931-5940 (2013).

Comment 3:

*3) Moreover, even if the presented scheme mitigates Q (at particular spectral regimes of mode coupling) by avoiding coupling to a higher-order mode in a limited case, its effect and scientific novelty are questionable. For example, a higher-order mode filtering, as demonstrated in *Opt. Lett.* **41**, 452 (2016), sounds more effective and generally applicable ways to avoid the coupling to a higher-order mode that the authors mainly claim in this paper.*

Reply:

We thank the reviewer for raising this point and providing us with the above for our reference. We agree with the reviewer that the demonstration in *Opt. Lett.* **41**, 452 (2016) is a useful way to induce a transition to a single-mode waveguide inside the multimode resonator for reduction of mode coupling, i.e., to induce a single-mode bend and loss that suppress the higher-order modes. The additional transition structures may introduce loss that results in a lower Q factor. The loaded Q factors in *Opt. Lett.* **41**, 452 (2016) have values from 2.4×10^5 to 9.7×10^5 from 1510 nm to 1610 nm. Other useful approaches, like the Euler bends and Bezier bends, have the same functions to suppress the higher-order modes.

For the Euler and Bezier bends, they follow the following equations to design^{42, 50}.

$$\frac{d\theta}{dL} = \frac{1}{R} = \frac{L}{A^2} + \frac{1}{R_{\max}} \tag{R3}$$

$$B(t) = P_0(1-t)^3 + 3P_1 \cdot t \cdot (1-t)^2 + 3P_2 \cdot t^2 \cdot (1-t) + P_3 \cdot t^3, t \in [0,1] \tag{R4}$$

In the above equations, L is the curve length; A is a constant related to the total length of the Euler bend; R_{\max} is the maximum radius of the Euler curve; B is the unitless number defining the curve radius change; P_0 and P_3 are the start and end points of the Bezier curve.

To further combine the above bends for broadband high-Q resonator, a directional coupler with a careful design is also required for comparable coupling coefficients at wavelength region over several hundreds of nanometers. Therefore, the corresponding directional couplers may need to be modified or other new structures would be required according to the unique shapes of different adiabatic bends.

Here, for the first time, we propose a new approach to using a multimode concentric resonator for the alleviation of the mode coupling from the higher-order modes. Since the inner resonator has a wider width than that of the outer resonator which means lower loss, the mutual coupling between them can result in an increased Q factor in the outer resonator. Even though more loss is introduced by coupling with the inner resonator, its Q factor is still maintained over 10^6 due to its low intrinsic loss. When the higher-order mode couples to one resonance and thus reduces its Q factor, another high-Q resonance is still present within that FSR. Besides, since the outer resonator is a multimode racetrack resonator, the pulley directional coupler is easy to be engineered. Therefore, using our proposed multimode concentric resonator combined with the pulley directional coupler, we are able to achieve the broadband high Q factors measured in a range of over 440 nm, where the measurement is limited by the range of our lasers. From the Raman laser measurement, the wavelength range for broadband high Q factors is expected to be over 600 nm since we achieved comparable Raman lasing output at the longer wavelengths with the comparable Q factors at the pump wavelengths. Using this multimode concentric resonator, we demonstrate a new record for the most widely tunable integrated Raman laser. We further compare our multimode concentric resonator with the multimode resonators using the above bends.

Table R1 | Comparison between multimode resonators using different bends to suppress the higher-order modes.

Material (work)	Bend type	Bus waveguide width (μm)	bend width (μm)	FSR	Q_L factor ($\times 10^6$)	Broadband high-Q wavelength range
Silicon nitride ⁴⁸	Single-mode bend	1.65	0.45	100 GHz	(2.4 ~ 9.7) $\times 10^5$	-
Silicon ⁴⁷	Single-mode bend	2	0.5	0.208 nm	1.1×10^6	-
Silicon ⁴²	Euler bend	0.45	1.6	0.9 nm	1.3×10^6	-
Silicon ⁴³	Euler bend	3	3	0.325 nm	9.4×10^6	-
Silicon nitride ⁴⁹	Euler bend	2.2	2.2	19.8 GHz	$\sim 10^7$	280
Silicon ⁵⁰	Bezier bend	0.45	0.45	0.036 nm	1.54×10^6	-
Silicon (this work)	Concentric bend	0.88	1.5 (outer); 3 (inner)	0.24 nm	1.4×10^6	> 600

-: not present in the paper.

We appreciate the reviewer for pointing out the above nice work for us to cite in our revised manuscript for readers' reference if they want to design a multimode bend. We also provided more references about the Euler bends and Bezier bends. They are described on page 3 in the revised manuscript as follows:

“This is a well-known problem and has been previously tackled using single-mode bends^{47, 48}, Euler bends^{42, 43, 49}, and Bezier bends⁵⁰ to suppress the higher-order modes at the bends and maintain a high Q

factor for the fundamental mode.”

42. Zhang L, Jie L, Zhang M, Wang Y, Xie Y, Shi Y, et al. Ultrahigh-Q silicon racetrack resonators. *Photonics Res.* **8**, 684-689 (2020).

43. Zhang L, Hong S, Wang Y, Yan H, Xie Y, Chen T, et al. Ultralow-loss silicon photonics beyond the singlemode regime. *Laser Photonics Rev.* 2100292 (2022).

47. Zhang Y, Hu X, Chen D, Wang L, Li M, Feng P, et al. Design and demonstration of ultra-high-Q silicon microring resonator based on a multi-mode ridge waveguide. *Opt. Lett.* **43**, 1586-1589 (2018).

48. Kordts A, Pfeiffer MH, Guo H, Brasch V, Kippenberg TJ. Higher order mode suppression in high-Q anomalous dispersion SiN microresonators for temporal dissipative Kerr soliton formation. *Opt. Lett.* **41**, 452-455 (2016).

49. Ji X, Liu J, He J, Wang RN, Qiu Z, Riemensberger J, et al. Compact, spatial-mode-interaction-free, ultralow-loss, nonlinear photonic integrated circuits. *Commun. Phys.* **5**, 1-9 (2022).

50. Mou B, Boxia Y, Yan Q, Yanwei W, Zhe H, Fan Y, et al. Ultrahigh Q SOI ring resonator with a strip waveguide. *Opt. Commun.* **505**, 127437 (2022).

Comment 4:

4) In addition, the authors presented temporal coupled modes for describing the mode couplings in spectral resonances. However, temporal coupled modes are an abstractive representation of the given phenomena plugging/fitting unknown parameters. It is an intuitive way to describe the couplings, but it does not allow ones to design or model the devices. Thus, I disagree with the authors' claim that they presented a new model to model/develop a resonator avoiding the coupling.

Reply: We thank the reviewer for the comment.

We agree with the reviewer that the temporal and spatial coupled-mode equations are well-established and intuitive models to describe mode interactions in coupled systems. Here, we use these two models to investigate and elaborate on how the multimode concentric resonators in both time and spatial domains help to achieve broadband high Q factors. The principle of the multimode concentric resonators is not to avoid coupling, but to have two high-Q modes (symmetric and anti-symmetric supermodes) in one FSR through mutual coupling with the inner resonator, such that even if one high-Q mode is destroyed by the higher-order mode, there remains another high-Q resonance for use. The two models well explain our theory and provide an intuitive understanding of the properties of the multimode concentric resonators. For the detailed design of the multimode concentric resonators, the widths of the outer and the inner resonators are first chosen to have equal optical path lengths at a design target wavelength, while the effective indices of the two resonators are obtained by FDTD simulation. For other wavelengths, despite their different optical length paths, Out_{TE0} and In_{TE0} can still couple to each other when both of them are on-resonance.

The approach of using the multimode concentric resonator to mitigate the mode coupling from the higher-order mode and thus keep at least one high-Q TE0 resonance is new. This new approach is well established by the well-known models of using the mode coupled equations to describe the coupling

characteristics.

To clarify our approach is a new approach to using the multimode concentric resonator to mitigate the mode coupling from the higher-order mode rather than a new model to develop the resonator, one related sentence is stated on page 3 of the revised paper as follows:

“Here, we propose a new approach using multimode concentric racetracks together with a broadband pulley design on the directional coupler region to maintain ultrahigh Q factors over a broadband wavelength range in the multimode silicon resonator as shown in Fig. 1a.”

Comment 5:

In my opinion, the presented work catches eyes at first glance and is well-studied work for a specific experiment. However, I disagree with the novelty and the significance that the authors claim and recommend this work be published in other journals.

Reply: We thank the reviewer for this comment. We are sorry that we did not present our work clearly to show the significance of our novel approach. We have tried to revise our presentation as follows:

We now start by introducing the common issue of mode coupling in the multimode resonator. And present some previous approaches like using single-mode bends, the Euler bends and Bezier bends to suppress the higher-order modes. Then, for the first time, we propose a new approach to using a multimode concentric bend that provides two high-Q resonances at one FSR. The mechanism is not to suppress the higher-order modes but to use one of the high-Q resonances to mitigate the mode coupling from the higher-order mode and keep at least one high-Q resonance that can be used in each FSR. Together with the pulley directional coupler, the whole multimode concentric resonator would be able to achieve broadband high Q factors with average loaded Q factors of 1.4×10^6 over a 440 nm wavelength range. The wavelength range for broadband high Q factors is expected to be over 600 nm as the obtained comparable Raman lasing output at the longer wavelengths beyond our laser wavelength range from the Raman laser measurements. We used the well-known coupled-mode equations in both time and spatial domains to explain this phenomenon. Both experiments and theoretical models are in good accordance.

As this new approach is not to suppress the higher-order modes but to alleviate the mode coupling, it may be possible to take advantage of the perturbations in dispersion near the mode coupling wavelengths. Even though the multimode resonator has normal dispersion, when mode and mutual coupling are present, it can locally modify the dispersion for the initialization of modulation instability which can be used for dark comb generation. That is, the multimode concentric resonators may be useful for other nonlinear applications such as comb generation or widely tunable parametric oscillators. The longest Raman lasing wavelength presented in this paper (1841 nm) was limited by the pump laser source and in principle, we can further extend the design for cascaded pumping to longer wavelengths. Generally, since the multimode concentric resonators address the problem of mode coupling with higher-order modes by

having at least one of the two resonances to maintain a high Q factor in every FSR, any applications requiring widely tunable wavelength range with low input power, broadband output, or individual high-Q resonances with large frequency detuning can find this approach useful.

Based on the reviewer's suggestions, we conducted measurements on multimode concentric resonators compared to multimode single resonators and compared them with the newly added numerical models, which make the manuscript more comprehensive. The experimental results well proved the functions of the multimode concentric resonators as a useful approach to mitigate the mode coupling between the higher-order modes and the fundamental modes which highly decreases the Q factors at the fundamental modes in the multimode single resonators. We believe that this revised manuscript has been significantly improved with novel results by containing experimental and numerical data. We hope that the reviewer finds our newly added results are of significance.

To emphasize the scientific innovation of our work novelty, we have added more descriptions of novelty by making the following changes, which have been underlined and marked with red color on page 13 of the revised paper as follows:

“The mechanism for maintaining broadband high Q factors is not to suppress the higher-order modes such as by using appropriately designed bends^{42, 47, 48, 50}, but to use one of the two non-degenerate high-Q resonances of the multimode concentric racetracks to mitigate the mode coupling of the other high-Q resonance with the higher-order mode. The outer and inner resonators are composed of multimode waveguides with normal dispersion. The pulley directional coupler can be easily engineered to have comparable coupling coefficients over a wide wavelength range. Thus, the multimode concentric resonators employed with the broadband pulley directional couplers can enable broadband high-Q resonances over several hundreds of nanometers wavelength ranges. This new approach does not suppress the higher-order modes but rather mitigates the reduction in Q factor within one FSR by providing an additional resonance near the mode coupling wavelengths. Even though the multimode resonator has normal dispersion, the presence of the mode and mutual coupling can modify the dispersion for the initialization of modulation instability which can be used for comb generation⁵³. The multimode concentric resonators may therefore find applications beyond that demonstrated in the widely tunable Raman laser and may be of interest for use in wideband comb generation⁶², or broadband integrated optical parametric oscillators⁶⁶.”

42. Zhang L, Jie L, Zhang M, Wang Y, Xie Y, Shi Y, *et al.* Ultrahigh-Q silicon racetrack resonators. *Photonics Res.* **8**, 684-689 (2020).

47. Zhang Y, Hu X, Chen D, Wang L, Li M, Feng P, *et al.* Design and demonstration of ultra-high-Q silicon microring resonator based on a multi-mode ridge waveguide. *Opt. Lett.* **43**, 1586-1589 (2018).

48. Kordts A, Pfeiffer MH, Guo H, Brasch V, Kippenberg TJ. Higher order mode suppression in high-Q anomalous dispersion SiN microresonators for temporal dissipative Kerr soliton formation. *Opt. Lett.* **41**, 452-455 (2016).

50. Mou B, Boxia Y, Yan Q, Yanwei W, Zhe H, Fan Y, *et al.* Ultrahigh Q SOI ring resonator with a strip waveguide. *Opt. Commun.* **505**, 127437 (2022).

53. Kim S, Han K, Wang C, Jaramillo-Villegas JA, Xue X, Bao C, *et al.* Dispersion engineering and frequency comb generation in thin silicon nitride concentric microresonators. *Nat. Commun.* **8**, 1-8 (2017).
62. Moille G, Perez EF, Stone JR, Rao A, Lu X, Rahman TS, *et al.* Ultra-broadband Kerr microcomb through soliton spectral translation. *Nat. Commun.* **12**, 1-9 (2021).
66. Kuyken B, Liu X, Osgood RM, Baets R, Roelkens G, Green WM. A silicon-based widely tunable short-wave infrared optical parametric oscillator. *Opt. Express* **21**, 5931-5940 (2013).

Comment 6:

Other comments:

- Instead of plotting the coupling fractions in Fig.1b, it would be better to plot in coupling-Q so that others can compare it with the coupling-Qs in Fig.2f.

Reply: Thanks for the reviewer’s suggestion. Based on the simulated coupling fraction κ^2 obtained by Lumerical 3D FDTD simulation, we calculated coupling Q factors⁵² using Equation R4.

$$Q_c = \frac{\omega}{c_0} \frac{2n_{g,s}L_s + 2\pi n_{g,bent}R_{out}}{|\kappa|^2} \quad (R4)$$

where ω is the resonant frequency of Out_{TE0}; c_0 is vacuum light speed; $n_{g,s}$ and $n_{g,bent}$ are the group indexes of the straight and bent parts of the outer racetrack; L_s is the length of the straight part of the outer racetrack; R_{out} is the radius of the outer racetrack. The experimental obtained coupling Q factors in Fig. 2f (Fig. R5b) are consistent with the simulated coupling Q factors with an average value of about 2.3×10^6 in Fig. R5a.

Fig. R5 | Coupling Q factors from simulation and experiment.

The coupling Q factors are plotted in Fig.1b replacing the previous coupling fractions to compare with the measured coupling Q factors in Fig.2f and the corresponding descriptions are added as follows:

“We first calculated the variation of coupling fractions of the directional coupler with wavelengths changing from 1200 nm to 1700 nm. The total coupling fraction fluctuates about an average value of 2.82%. The coupling fraction from the bus waveguide to the TE0 mode of the outer racetrack has a 1 dB bandwidth of about 283 nm at wavelengths from 1300 nm to 1583 nm. The corresponding coupling Q factors⁵² can be calculated with an average value of 2.3×10^6 in Fig. 1b.”

52. Moille G, Li Q, Briles TC, Yu S-P, Drake T, Lu X, *et al.* Broadband resonator-waveguide coupling for efficient extraction of octave-spanning microcombs. *Opt. Lett.* **44**, 4737-4740 (2019).

Fig. 1 | The concentric racetrack resonator design. **a**, Structure of the concentric racetrack resonator along with a p-i-n junction to remove the free carriers. Schematic from the top view with geometric parameters. w_0 : $0.88 \mu\text{m}$; w_{in} : $3 \mu\text{m}$; w_{out} : $1.5 \mu\text{m}$; gap1 : $0.4 \mu\text{m}$; gap2 : $0.4 \mu\text{m}$; R_{out} : $200 \mu\text{m}$; L_s : $690 \mu\text{m}$. **b**, Simulated coupling Q factors versus wavelengths from $1200 - 1700 \text{ nm}$. **c**, Simulated mode propagates mainly in the outer racetrack, and modes are excited in the inner racetrack. **d**, Calculated optical path lengths at the fundamental transverse electric (TE₀) modes of the outer and the inner racetracks. **e**, Simulated transmission spectra around 1550 nm with different coupling coefficients u between the inner and outer racetracks. **f**, Simulated transmission spectrum of the multimode concentric resonator from 1480 nm to 1481.5 nm .

Comment 7:

- Fig.1f: y-axis label is recommended to be changed.

Reply: Thanks for the reviewer's suggestion.

The y axis of Fig. 1f has been changed to normalized transmission in the revised manuscript as follows:

Thanks for your valuable time. We greatly appreciate the comments which help to make the paper better.

Yours sincerely,

Dr. Yaojing Zhang, Ms. Keyi Zhong, Mr. Xuetong Zhou, and Dr. Hon Ki Tsang

Center for Advanced Research in Photonics

Department of Electronic Engineering

The Chinese University of Hong Kong, Shatin, Hong Kong

Email: hktsang@ee.cuhk.edu.hk

REVIEWERS' COMMENTS

Reviewer #1 (Remarks to the Author):

I appreciate the authors for the rigorousness in their response letter, especially for re-highlighting the significance/novelty of this work. It's still questionable for the generalization of this approach to other platforms and its applicability, but this would be beyond the scope of this paper. Given the revision, I recommend this manuscript be published in Nature Communications.